# An optical neural chip for implementing complex-valued neural network

H. Zhang[1], M. Gu [2,3✉], X. D. Jiang [1✉], J. Thompson [3], H. Cai[4], S. Paesani[5], R. Santagati [5], A. Laing [5], Y. Zhang [1,6], M. H. Yung[7,8], Y. Z. Shi [1], F. K. Muhammad[1], G. Q. Lo[9], X. S. Luo [9], B. Dong[9], D. L. Kwong[4], L. C. Kwek [1,3,10✉] & A. Q. Liu [1✉]

Complex-valued neural networks have many advantages over their real-valued counterparts. Conventional digital electronic computing platforms are incapable of executing truly complex-valued representations and operations. In contrast, optical computing platforms that encode information in both phase and magnitude can execute complex arithmetic by optical interference, offering significantly enhanced computational speed and energy efficiency. However, to date, most demonstrations of optical neural networks still only utilize conventional real-valued frameworks that are designed for digital computers, forfeiting many of the advantages of optical computing such as efficient complex-valued operations. In this article, we highlight an optical neural chip (ONC) that implements truly complex-valued neural networks. We benchmark the performance of our complex-valued ONC in four settings: simple Boolean tasks, species classification of an *Iris* dataset, classifying nonlinear datasets (Circle and Spiral), and handwriting recognition. Strong learning capabilities (i.e., high accuracy, fast convergence and the capability to construct nonlinear decision boundaries) are achieved by our complex-valued ONC compared to its real-valued counterpart.

[1] Quantum Science and Engineering Centre (QSec), Nanyang Technological University, 50 Nanyang Ave, 639798 Singapore, Singapore. [2] Complexity Institute and School of Physical and Mathematical Sciences, Nanyang Technological University, 50 Nanyang Ave, 639798 Singapore, Singapore. [3] Centre for Quantum Technologies, National University of Singapore, Block S15, 3 Science Drive 2, Singapore 117543, Singapore. [4] Institute of Microelectronics, A*STAR (Agency for Science, Technology and Research), 138634 Singapore, Singapore. [5] Centre for Quantum Photonics, H. H. Wills Physics Laboratory and Department of Electrical and Electronic Engineering, University of Bristol, Merchant Venturers Building, Woodland Road, Bristol BS8 1UB, UK. [6] School of Mechanical & Aerospace Engineering, Nanyang Technological University, 50 Nanyang Ave, 639798 Singapore, Singapore. [7] Institute for Quantum Science and Engineering, Department of Physics, Southern University of Science and Technology, Shenzhen 518055, China. [8] Shenzhen Key Laboratory of Quantum Science and Engineering, Southern University of Science and Technology, Shenzhen 518055, China. [9] Advanced Micro Foundry, 11 Science Park Road, 117685 Singapore, Singapore. [10] National Institute of Education, 1 Nanyang Walk, 637616 Singapore, Singapore. ✉email: gumile@ntu.edu.sg; exdjiang@ntu.edu.sg; cqtklc@nus.edu.sg; eaqliu@ntu.edu.sg

Advanced machine learning algorithms, such as artificial neural networks[1,2], have received significant attention for their potential applications in key tasks such as image recognition and language processing[3–5]. Notably, neural networks make heavy use of multiply-accumulate (MAC) operations, causing heavy computation burden in existing electronic computing hardware (e.g., CPU, GPU, FPGA, ASIC). Application-specific devices for executing MAC operations are preferred. Currently, the vast majority of existing neural networks rely entirely on real-valued arithmetic, whereas complex arithmetic may offer a significant advantage. For instance, the detection of symmetry problem and XOR problem can be easily solved by a single complex-valued neuron with orthogonal decision boundaries, but cannot be done with a single real-valued neuron[6]. Meanwhile, recent studies suggest that complex-valued arithmetic[7,8] would significantly improve the performance of neural networks by offering rich representational capacity[9], fast convergence[10], strong generalization[11] and noise-robust memory mechanisms[12]. Conventional digital electronic computing platforms exhibit significant slowdown when executing algorithms using complex-valued operations because complex numbers have to be represented by two real numbers[7,13], which increases the number of MAC operations—the most frequently used and computationally expensive component of the neural network algorithms[14,15]. To overcome these hurdles, it has been proposed that the computationally taxing task of implementing neural networks be outsourced to optical computing[16] which is capable of truly complex-valued arithmetic.

Optical computing offers advantages like low power consumption[17,18], high computational speed[19], large information storage[20], inherent parallelism[21] that cannot be rivaled by its electronic counterpart. Several optical implementations of neural networks have been proposed. Among these technologies, photonic chip-based optical neural networks have become increasingly mainstream for their high compatibility, scalability, and stability. This platform has already achieved notable success in demonstrating neuromorphic photonic weight banks[22–24], all-optical neural networks[25,26] and optical reservoir computing[27,28]. A classical fully connected neural network has been experimentally demonstrated on an integrated silicon photonic chip[29,30]. Although this optical chip is based on light interference, the implemented neural network algorithms are real-valued, which forfeits the benefits of complex-valued neural networks. A highly parallelized optical neural network accelerator based on photo-electric multiplication has been reported[31,32], which is also designed for real arithmetic because the optical signals had already been converted to photocurrents before reaching the accumulator. Other topics related to optical neural networks include on-chip training[33], optical nonlinear activations[34–36] and various neural network architectures[37–39].

Besides optical computing platforms, analogue electronic devices, as opposed to the more mainstream digital electronic devices, have successfully demonstrated multilayer perceptrons[40,41] and convolutional neural networks[42]. Complex-valued neural networks on analogue electronic devices have already been explored in some previous works[43–45]. In reservoir computing, complex-valued reservoirs also contribute to enriched system dynamics and improved performance[46,47]. However, few explorations have been made in optical computing platforms for implementing general-purpose and complex-valued neural networks despite the fact that the optical neural networks are able to process information in multiple degrees of freedom (e.g., magnitude and phase) by complex-valued arithmetic and obtain more efficient information processing and analysis[48,49]. Existing optical implementations have not stepped into this potential flatland due to their reliance on classical deep learning algorithms designed for real-valued arithmetic on conventional electronic computers. These real-valued optical neural networks are implemented solely using the intensity information of the optical signals while discarding the phase information, which forfeits a key benefit of optical computing.

We tackle these issues by proposing and experimentally realizing an optical neural chip (ONC) that executes complex-valued arithmetic, highlighting the advantages of chip-based complex-valued networks by optical computing. This results in a complex-valued neural network that integrates input preparation, weight multiplication and output generation in a single photonic chip. Meanwhile, previous complications of complex-valued networks—cumbersome arithmetic on complex numbers—are alleviated by directly realizing such operations through optical interference. We experimentally benchmark our complex-valued ONC in multiple practical settings including (a) realization of elementary logic gates, (b) classification of *Iris* species, (c) classification of nonlinear datasets (i.e., Circle and Spiral) and (d) handwriting recognition using a multilayer perceptron (MLP) network, and compare its performance with a similar on-chip implementation using real-valued perceptrons. In all cases, our complex-valued ONC demonstrates remarkable performance. In the elementary gate realization, we illustrate the realization of several logic gates including a nonlinear XOR gate by a single complex-valued neuron—a task which is impossible for a single real-valued neuron. In *Iris* classification, we obtain an accuracy of up to 97.4% in chip testing. The nonlinear decision boundaries are visualized in the classification of Circle and Spiral datasets. In the handwriting recognition task, we achieve a testing accuracy of 90.5% using a $4 \times 4$ hidden layer, being an 8.5% improvement over the real-valued counterpart. Moreover, the performance gap persists when the encoding and decoding modules are in intensity only—indicating that tour phase-sensitive ONC exhibits operational advantage even for all real-valued interfaces. Our results present a promising avenue towards realizing deep complex-valued neural networks with dedicated integrated optical computing chips, and potential implementations of high dimensional quantum neural networks.

## Results

**Design and fabrication**. Figure 1a shows the architecture of the optical neural network, which is composed of an input layer, multiple hidden layers and an output layer. In the complex-valued architecture, light signals are encoded and manipulated by both optical magnitude and phase during the initial input signal preparation and network evolution. Figure 1b shows the schematic of the ONC to implement complex-valued neural networks. The input preparation, weight multiplication and coherent detection are all integrated onto a single chip. A coherent laser (wavelength 1550 nm) is used to generate the input signals. The ONC is essentially a multiport interferometer, in which Mach–Zehnder interferometers (MZIs) are arranged in a specific manner[50–52]. Each MZI consists of two beam splitter (BS)–phase shifter (PS) pairs. The transmissivity of the BS is fixed at 50:50, and the PS is thermally modulated to tune the phase. In the diagram, MZIs marked with different colours have different functionalities. The coherent laser is coupled into the chip from the bottom port. Input light division and modulation are realized by the chain of MZIs marked in red. The green marked MZI separates the reference light that will be used for coherent detection. The on-chip light division makes sure that the light signals propagating along different optical paths have the same polarization and share a stable relative phase. The input modulation is dictated by the machine learning task. For tasks with real-valued inputs, the light signals are modulated by the magnitude, and the relative phases between different paths are set to zero. For complex-valued inputs, the modulation includes both magnitude modulation and path-dependent phase rotations. All

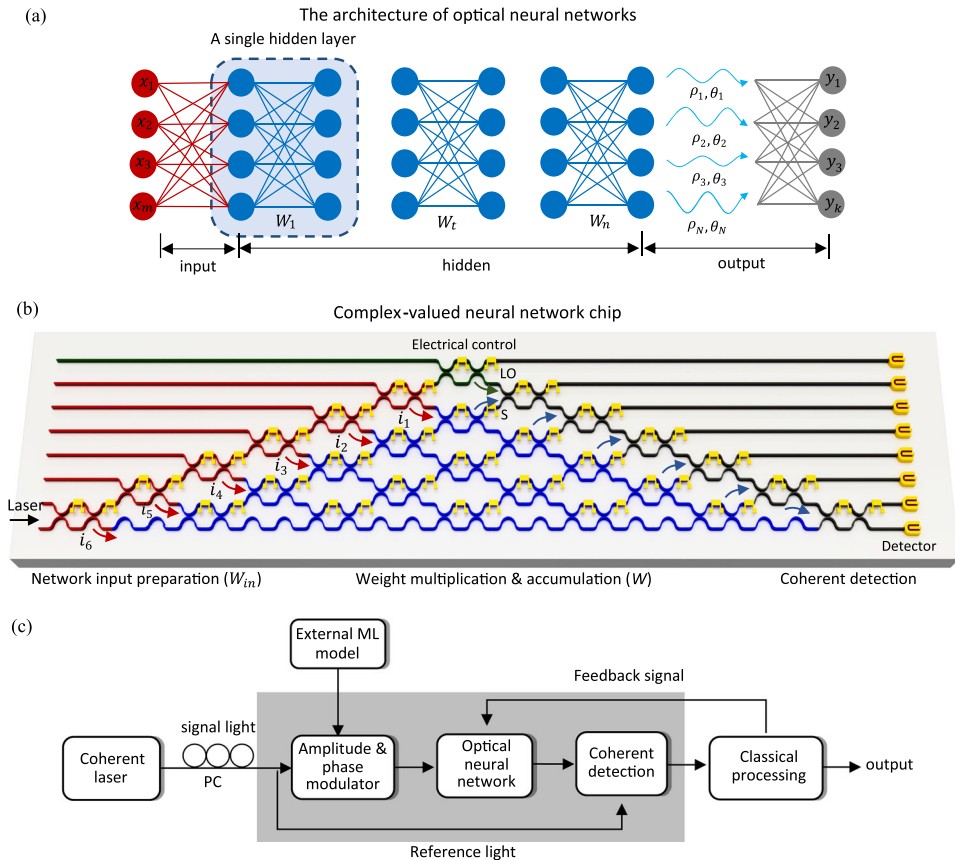

**Fig. 1 The composition of complex-valued coherent optical neural network. a** An optical neural network is composed of an input layer, multiple hidden layers and an output layer. In our complex-valued design, the light signals are encoded and manipulated by both magnitude and phase during the initial input preparation and network evolution. **b** The schematic of the ONC in implementing complex-valued networks. The input preparation, weight multiplication and coherent detection are all integrated onto a single chip. The division and modulation of the light signals ($i_1$–$i_6$) are realized by the MZIs marked in red. The green marked MZI separates the reference light that will later be used for coherent detection. The MZIs used to implement the 6 × 6 complex-valued weight matrix are marked in blue. The remaining grey marked MZIs are used for on-chip coherent detection. **c** The workflow of the ONC system. A coherent laser at 1550 nm is used to generate signal light and reference light. The signal light on each path is modulated by its magnitude and phase according to the machine learning (ML) task. The weighted sum operation is accomplished passively through light inference. The measurement results are sent to the electrical interface for processing, including the application of activation function and the calculation of cost function. The ONC chip are then reconfigured accordingly by the updated weight matrices.

light signals, as well as the reference light, are generated on chip from a single coherent laser and are modulated by the same chain of PSs. Stringent control is required over the phases of the light signals, when implementing either complex-valued or real-valued networks on the coherent chip. The integration of the light division and modulation effectively avoids the possible phase fluctuations which take place when coupling external light signals to the chip.

After the input preparation, six light signals and a reference light are available. Then, the light signals travel through the 6 × 6 optical neural network marked blue in Fig. 1b. An $N$-mode network realizes the weight matrix multiplication by transforming the input states into output states according to $S_{out} = U(N)S_{in}$. $U(N)$ is a $N \times N$ unitary matrix that represents the product of multiple rotation matrices $\{T_{pq}\}$ and a diagonal matrix $D$, such that $U(N) = \prod_{p=2}^{N}\prod_{q=1}^{p-1} T_{pq}D$, where the modulus of complex elements on the diagonal of $D$ equal to one, and $T_{pq}$ is defined as the $N$-dimensional identity matrix with the elements $t_{pp}$, $t_{pq}$, $t_{qp}$ and $t_{qq}$ replaced by

$$\begin{bmatrix} t_{pp} & t_{pq} \\ t_{qp} & t_{qq} \end{bmatrix} = \mathrm{i}e^{\mathrm{i}\frac{\theta}{2}} \begin{bmatrix} e^{\mathrm{i}\phi}\sin\frac{\theta}{2} & e^{\mathrm{i}\phi}\cos\frac{\theta}{2} \\ \cos\frac{\theta}{2} & -\sin\frac{\theta}{2} \end{bmatrix} \quad (1)$$

where $\theta$ is defined as the internal PS between two BSs and $\phi$ is the external PS. An optical network with $N$ inputs realizes an arbitrary $N \times N$ unitary weight matrices $U(N)$ by adjusting the tuneable PSs on the MZIs. Detection-based implementation of activation functions are adopted in our demonstrations.

The MZIs marked in grey are used for on-chip coherent detection. The output light signals of the optical chip contain information in both magnitude and phase, while conventional intensity detection techniques only access magnitude information. Our integrated chip is capable of both intensity and coherent detection. The goal of coherent detection is to determine the phase angle $\phi_s$ between the reference light and signal light. By connecting photodiodes at both outputs in a balanced way, the obtained output current is $I_I \propto 2A_sA_1\cos\phi_s$, where $A_s$ and $A_1$ are the respective magnitudes of the signal and the reference light. Similarly, by adding a phase shift of $\pi/2$ to the reference light, the output current is $I_Q \propto 2A_sA_1\sin\phi_s$. The $\phi_s$ is then determined from the ratio of $I_I$ and $I_Q$, which also helps eliminate the physical noise from the optical components. The choice of detection method is determined by the activation function. Intensity detection is naturally adopted for the activation function $M(z) = \|z\|$, meanwhile the coherent detection is adopted for the activation function $\mathrm{ModReLU}(z) = \mathrm{ReLU}(\|z\| + b)e^{\mathrm{i}\theta_z}$. The

detected photocurrents are converted into voltage signals by a transimpedance amplifier (TIA), and then collected and processed by a classical processor with an analogue-to-digital converter (DAC). Feedback signals can be generated and sent back to the ONC to adjust the chip configurations as shown in Fig. 1c.

The packaged ONC for a complex-valued neural network is shown in Fig. 2a. The false-colour micrographs of the optical

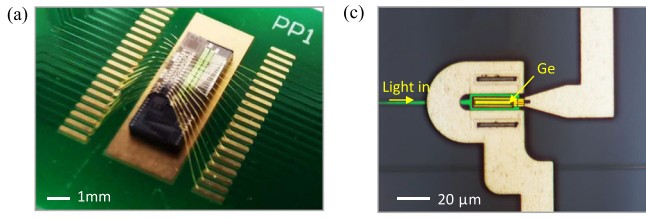

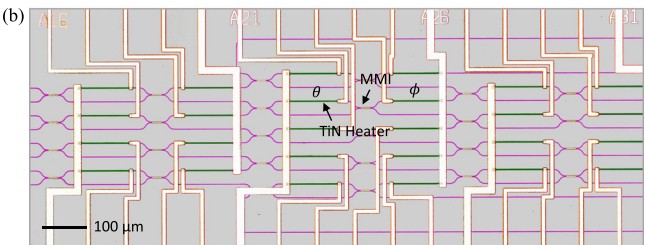

**Fig. 2 Fabrication and packaging of silicon photonic chip. a** Chip packaging. **b** A false-colour micrograph of the the MZI network with integrated heaters. **c** A false-colour micrograph of an on-chip photodetector.

network and the waveguide-coupled Ge-on-SOI photodetector are shown in Fig. 2b, c, respectively. In this work, an ONC with 8 modes and 56 PSs is used. The internal PS $\theta$ and external PS $\phi$ of each MZI are as marked. The 50:50 BS is realized by a multimode interference (MMI) device. All the PSs are thermally tuned with integrated titanium nitride (TiN) heaters bonded to a PCB. The heaters are calibrated and fitted with an average $R$-square value of 0.99 (see Supplementary Fig. 5). The decomposition of a complex-valued matrix follows the MZI arrangement as shown in Fig. 1b, and each MZI is described by Eq. (1). Also refer to Supplementary Notes 4–6, including the input encoding process, the decomposition and implementation of the weight matrices on chip. Coherent detection is conducted on chip (see Supplementary Note 7).

**Neuron model and task briefing**. The complex-valued neuron mirrors the conventional neuron model, where all parameters and variables are complex-valued, and the computation employs complex-valued arithmetic. The neuron is built by weighting each input with a complex number, as shown in Fig. 3a. The weighted inputs are summed up and processed by an activation function. The output of the neuron is expressed as

$$y = f\left(\sum_{i=1}^{n} w_i x_i + b\right) \quad (2)$$

where the weights $w_i$ and bias $b$ are in general complex numbers. Each input $x_i$ to the neuron can either be complex-valued or real-valued.

We benchmark our ONC implementation of such complex-valued computations in several separate tasks, and compare them

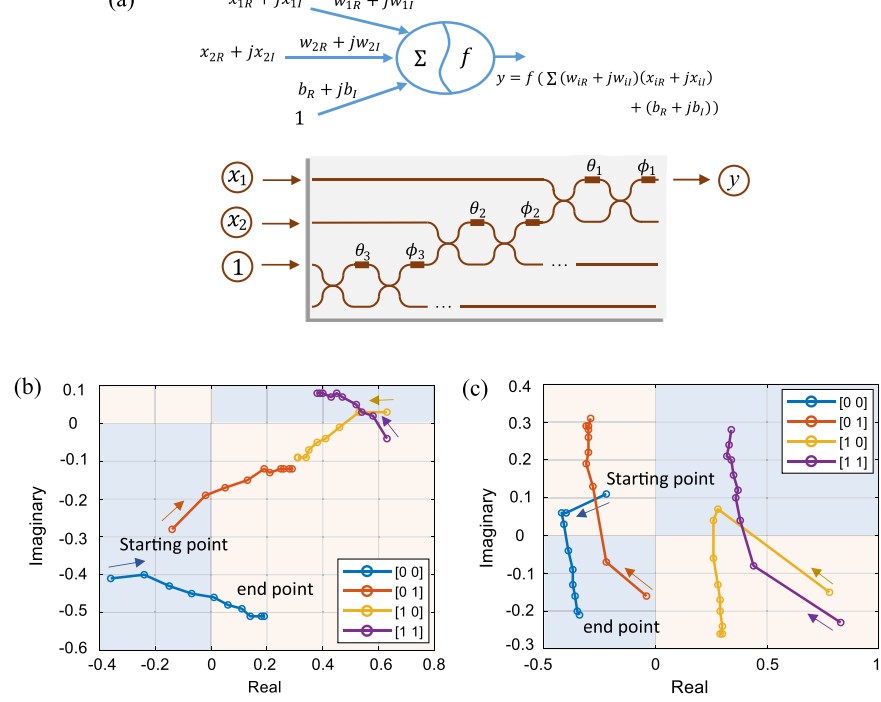

**Fig. 3 A single complex-valued neuron for logic gate task. a** The diagram of a complex-valued neuron and its on-chip implementation. The neuron output is the weighted sum of the input (e.g., $x_{1R} + jx_{1I}$, where the footnote R/I represents the real and imaginary part, respectively) and complex weights (e.g., $w_{1R} + jw_{1I}$), after being processed by an activation function $f$. The bias is implemented with an additional constant input weighted by a trainable complex-valued weight (i.e., $b_R + jb_I$). The weight matrix is decomposed to the phase shift values ($\theta_i$ and $\phi_i$) on chip. **b** The training process of NAND gate and **c** The training process of XOR gate. 10 iterations are conducted and recorded for each logic gate. The quadrants representing logical 0 are painted blue and those representing logical 1 are painted pink. Being processed by a complex-valued neuron, each of the four possible combinations of logical inputs converges from a random starting point to the expected end point, via a continuous modulation of magnitude and phase rotation.

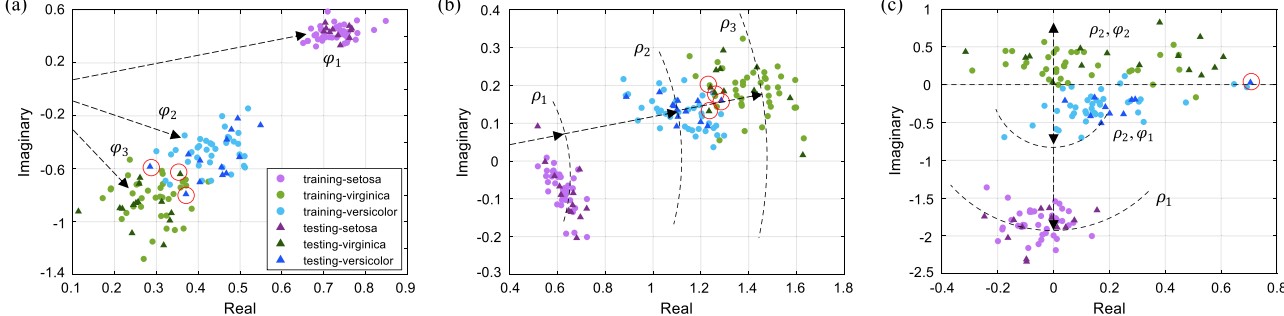

**Fig. 4 A complex-valued layer for classification of dataset *Iris*.** The ONC is reconfigured with the complex-valued weight matrix trained for *Iris* task. The complex-valued neuron outputs are measured and displayed in two-dimensional Cartesian diagrams, as in **a–c**. The three shown cases are trained by three respective targets: phase only ($\varphi_1$, $\varphi_2$, $\varphi_3$), magnitude only ($\rho_1$, $\rho_2$, $\rho_3$) and their combinations ($\rho_1$, ($\rho_2$, $\varphi_1$), ($\rho_2$, $\varphi_2$)). The circle markers are the validation results of training dataset while the triangle markers are those of the blind test instances. Data points which were wrongly predicted in the blind test are circled in red. It is shown by validation results that the optical chip identifies successfully the three species from the known data. While from testing results, we learn that the generalized model also fits well to unknown data.

against a similarly configured optical chip that computes only on real values:

(a) The implementation of fundamental two-bit logic gates using a single complex neuron. Notably, this includes XOR gate, which cannot be accomplished by a real-valued neuron, and usually requires a three-layered real-valued neural network[53].

(b) The use of a single complex layer to classify *Iris*[54,55] dataset into three subspecies, benchmarking against a three-layer real network[56,57]. The complex layer achieves a high accuracy of 97.4% with a single layer, thereby reducing cumulative errors.

(c) The classification of the nonlinear dataset Circle and Spiral by simple complex networks, with visualization of the decision boundaries. The complex networks achieve significantly high accuracy, which is unmatched by similar real networks, showing its strong capability in learning nonlinear patterns.

(d) The task of handwriting recognition using a complex MLP configured on the ONC. Our benchmarks based on the MNIST database[58] illustrate that the complex-valued network attains a much higher accuracy (90.5% vs. 82.0%), even when the neural chip was only given real-valued inputs and constrained to deliver real valued outputs (86.5% *vs.* 82.0%).

**Logic gate realization**. The logic gate task provides a illustrative example, in which we emulate the implementation of logic gates by complex-valued networks to showcase the capability of the basic unit (a single complex-valued neuron) of our device. When implementing fundamental logic gates, the inputs to a neuron are real values, which can also be regarded as complex values whose phases are constrained to 0 or $\pi$. The bias is implemented by a constant input 1, and an additional complex-valued variable $b$ is allocated in the weight vector to make the bias trainable. In this task, identity activation function (Supplementary Note 14) is applied to the single neuron. Therefore, Eq. (2) is simplified to $y = \mathbf{x} \cdot \mathbf{W}$, where the input vector $\mathbf{x} = [x_1, x_2, \ldots, x_n, 1] \in \mathbb{R}^{1 \times (n+1)}$ and the weight vector $\mathbf{W} = [w_1, w_2, \ldots, w_n, b]^{\mathrm{T}} \in \mathbb{C}^{(n+1) \times 1}$. The weight vector is updated after each iteration by $\mathbf{W} \leftarrow \mathbf{W} + \Delta\mathbf{W}$, where $\Delta\mathbf{W} = \eta(\hat{y} - y) \times \mathbf{x}^{\mathrm{T}}$, $\eta$ is the learning rate, $\hat{y}$ is the expected output and $y$ is the actual output. For all logic tasks, the mapping from complex-valued output to logical value is predefined as odd quadrants to logical value 0 and even quadrants to logical value 1. Figure 3a shows the schematic of the on-chip implementation of a complex-valued neuron. The three inputs to the neuron are encoded by magnitude, two of which are used for the logical inputs, and the remaining one is a constant 1 for bias. The weight vector $\mathbf{W} \in \mathbb{C}^{3 \times 1}$ is realized on chip by configuring three MZIs containing six adjustable parameters $\{\theta_i, \phi_i\}$, $i = 1, 2, 3$. Coherent detection is applied to obtain both the magnitude and phase of the output light signal.

Figure 3b, c show the training process of the NAND and XOR gates, respectively (see also Supplementary Fig. 10 for AND and OR gates). A total of 10 iterations are conducted and recorded for each logic gate. Quadrants that represent logical value 0 are painted in blue and those represent logical value 1 are painted pink. In a complex-valued neuron, each of the four possible combinations of logical inputs converges from a random starting point to the expected end point, through continuous magnitude modulation and phase rotation. Convergence is also observed in the arithmetic loss between the expected end point and the prediction result of the complex neuron, in both real and imaginary parts (Supplementary Fig. 11). Meanwhile, the final classification results are consistent with the truth tables (Supplementary Table 2). The complex-valued neuron can also be applied to solve general XOR problem (Supplementary Fig. 12). By using a single complex-valued neuron on ONC to realize logic gates, we demonstrate its ability to solve linear tasks, as well as certain tasks that are linearly inseparable in the real domain like the XOR gate. Notably, these tasks aim to reveal the qualitative differences in learning capability between a complex and a real neuron, rather than the physical realization of logic gates. The XOR gate represents a pattern that a complex neuron can recognize but a real neuron cannot. Even in the simplest case of a single neuron, the complex neuron exhibits richer behaviour than its real counterpart. Suppose that we are provided with a fixed number of neurons, the complex network could identify more sophisticated patterns than its real counterpart.

**Classification of dataset *Iris***. Our second benchmark of ONC is *Iris* classification by a single complex layer. Here, the task is to classify a given *Iris* flower into one of the three possible subspecies (*setosa, versicolor* and *virginica*) based on four real-valued inputs (the length and width of the petals and sepals). The non-triviality of this task is that the three species are indistinguishable by any single one of the four features. The overlaps between features of the three subspecies are shown in Supplementary Fig. 13.

The entire dataset with 150 instances is split into training set and testing set by a ratio of 0.75:0.25. The weights are trained only on the training set, based on the same numerical model as in the logic gate task. In addition to the four inputs, the bias is included

in the input vector as an entry. An additional variable is included in the weight matrix to make the bias trainable. In chip implementation, the real-valued input vectors are encoded by the magnitude of the light signals, while keeping their phases identical. The outputs are complex values and are acquired by coherent detection. The input vectors and trained weights are numerically decomposed into exact phase shift values (Supplementary Notes 5 and 6). The electric power required by each PS is calculated by its calibration curve (Supplementary Fig. 5). In this way, we configure our ONC with trained weights and show the neuron outputs in Fig. 4a–c. Here, the three cases shown are trained using three respective targets: phase only ($\varphi_1$, $\varphi_2$, $\varphi_3$), magnitude only ($\rho_1$, $\rho_2$, $\rho_3$) and their combinations ($\rho_1$, ($\rho_2$, $\varphi_1$), ($\rho_2$, $\varphi_2$)). From the output distribution in the complex plane, it can be intuitively observed that instances from the same subspecies are distributed in clusters. Both the training set and the testing set are evaluated on-chip, in order to validate the training process as well as to demonstrate the generalization capability of the trained model. In the figures, the validation results (on training set) are displayed by coloured circles, while the blind test results (on testing set) are displayed by triangles. Different colours represent different subspecies. The validation results show that the optical chip successfully classifies the three species on known data, while the testing results prove that the generalized model also fits well to unknown data. The species *setosa* is clearly distinguished from the rest, and the species *versicolor* and *virginica* are separated into two clusters with marginal overlap. Instances that are misclassified in the blind test are circled in red. The accuracies of the three blind tests are 92.1%, 89.5% and 97.4%, respectively.

We benchmark the single complex layer against three-layer real network that is commonly required for *Iris* classification in other photonic implementations of neural networks[58,59]. Our complex layer has a simulated accuracy of 99.3% (Supplementary Figs. 14 and 15) and a chip testing accuracy of 97.4%. Whereas for the three-layer real network, despite its comparable simulated accuracy of 97.3%, would experience a large decrease in accuracy due to the multilayer cumulative error during physical implementation (Supplementary Fig. 16). Although the improvement in simulated accuracy is not evident, our complex model obtains high accuracy in physical implementation using fewer layers and neurons, thereby avoiding excessive cumulative errors.

**Classification of nonlinear datasets Circle and Spiral**. Here, we highlight the capability of complex networks in forming nonlinear decision boundaries, in comparison to their real counterparts. Two nonlinear datasets are studied, namely the Circle and the Spiral. The dataset visualization, model construction (real/complex-valued for comparison) and chip measurement results are shown from left to right in Fig. 5a. Both datasets are entangled and linearly inseparable. They have two real-valued inputs and two classification classes. For each task, a complex model and an equivalent (same number of layers and neurons) real model are compared. For classification of the Circle, a single complex/real layer with two neurons is adopted. Intensity detection is performed at the output ports of the chip. For the Spiral, a two-layer network is adopted. The first layer has four neurons and are designed as complex-/real-valued for comparison. Intensity detection is performed at this layer, which is equivalent to the application of the activation function $M(z) = ||z||$. The second layer is a real-valued linear mapping from the four hidden nodes to the two output nodes. The classification results can be interpreted from the chip outputs $y_1$, $y_2$ by a simple manner of *Argmax*: if $y_1 \geq y_2$, the corresponding instance belongs to the class blue, otherwise if $y_1 < y_2$, it belongs to the class pink.

In the binary classification, the decision boundary partitions the underlying vector space into two regions, one for each class. The decision boundaries of the simulated real-valued model, the complex-valued model and the on-chip implemented complex model when classifying the Circle and the Spiral are shown in Fig. 5b, c, respectively. As shown, the decision boundaries of the real model are formed by straight lines, while the decision boundaries of the complex model are nonlinear curves that perfectly match the entangled shape of the datasets. The complex model is also appreciably superior in classification accuracy. The complex model achieves simulated accuracy of 100% on both datasets, far exceeding the 55% (on Circle) and 89% (on Spiral) achieved by the real model. In chip implementation, we scan both inputs $x_1$ and $x_2$ from −1 to 1 by a step size of 0.1. A total 441 input sets are tested. In experiment results, the black wires are our expected decision boundaries while the white wires are experimental ones, from which we can easily figure out which points are incorrectly classified. The chip testing accuracies are 98% for the Circle and 95% for the Spiral. The theoretical decision boundaries of the complex model are smooth, while the visualized resolution is reduced by the input interval in experiment.

The theoretical decision boundaries can be derived. Suppose we have a single layer with two neurons, the outputs of the real model are

$$\begin{bmatrix} y_1 \\ y_2 \end{bmatrix} = \begin{bmatrix} w_{11} & w_{12} \\ w_{21} & w_{22} \end{bmatrix} \begin{bmatrix} x_1 \\ x_2 \end{bmatrix} + \begin{bmatrix} b_1 \\ b_2 \end{bmatrix} \tag{3}$$

where the $w_{11,12,21,22}$ and $b_{1,2}$ are real-valued, $x_{1,2}$ are real inputs and $y_{1,2}$ are the outputs. The decision boundary is derived by solving the equation $||y_1|| = ||y_2||$. Therefore, the decision surface is formed by two straight lines:

$$\begin{cases} l_1 : (w_{11} - w_{21})x_1 + (w_{12} - w_{22})x_2 + (b_1 - b_2) = 0 \\ l_2 : (w_{11} + w_{21})x_1 + (w_{12} + w_{22})x_2 + (b_1 + b_2) = 0 \end{cases} \tag{4}$$

In the complex model, we replace the weight matrices by $w_{jk} = p_{jk} + iq_{jk}$, and $b_j = m_j + in_j$, where $j$, $k$ = 1, 2. By solving $||y_1|| = ||y_2||$, a nonlinear decision boundary is formed by

$$\begin{aligned} (p_{11}x_1 + p_{12}x_2 + m_1)^2 &+ (q_{11}x_1 + q_{12}x_2 + n_1)^2 \\ = (p_{21}x_1 + p_{22}x_2 + m_2)^2 &+ (q_{21}x_1 + q_{22}x_2 + n_2)^2 \end{aligned} \tag{5}$$

which can be simplified to a binary quadratic equation:

$$Ax_1^2 + Bx_2^2 + Cx_1x_2 + Dx_1 + Ex_2 + F = 0 \tag{6}$$

Equation (5) can form various curves, such as parabola, circle, ellipse, and hyperbola, with different parameters A–F, which can be learned from training data. For nonlinear dataset, the complex-valued network shows strong learning capability, by forming nonlinear decision boundaries and achieving high classification accuracy.

**Handwriting recognition by a complex-valued multilayer perceptron**. A single complex layer implemented on ONC is employed to build a multilayer perceptron (MLP) to classify handwritten digits in the dataset MNIST. The dataset is split into training and testing sets. Our model is trained on the entire training set, and 200 instances in the testing set are used to validate the trained model on-chip. As shown in Fig. 6a, the network consists of an input layer $W^{in}$, a hidden layer $W$ and an output layer $W^{out}$. The neuron numbers in the three layers are 4, 4 and 10, respectively. The input $28 \times 28$ grayscale image is reshaped into a $784 \times 1$ vector and compressed by the input layer into four features, which are then fed to the $4 \times 4$ hidden layer. The output layer maps the four hidden outputs to 10 classes, representing digits from 0 to 9. The simulation model is built in TensorFlow and trained by RMSPropOptimizer, with a learning

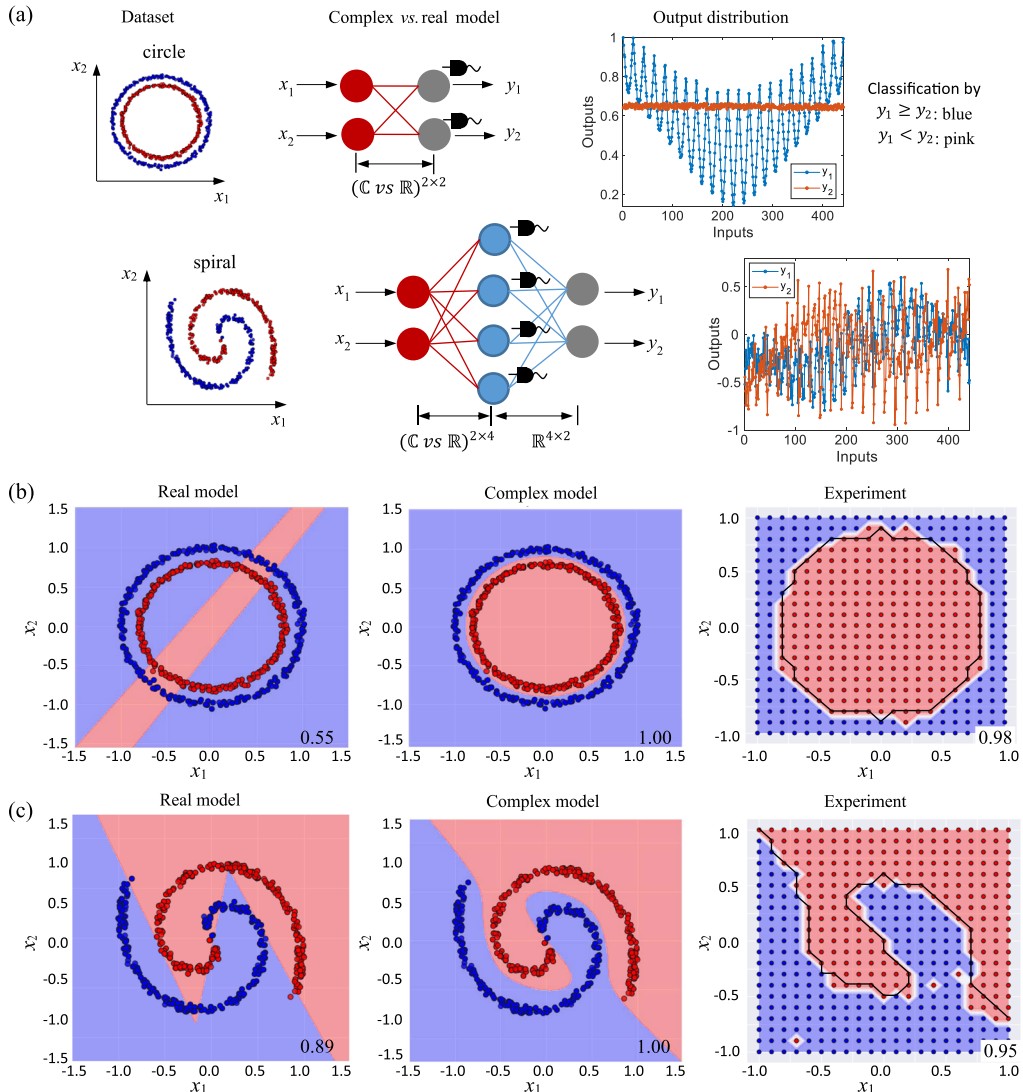

**Fig. 5 Nonlinear decision boundaries by complex-valued networks. a** Displayed from left to right are the dataset visualization, model construction and measured chip outputs. Two nonlinear datasets, the Circle and the Spiral, are investigated. Both datasets have two real-valued inputs and two classes. For classification of the Circle, a single layer (designed as complex vs. real for comparison) with two neurons is used. Intensity detection (represented by the PDs) is performed to the chip outputs. For the Spiral, a two-layer network is adopted. The first layer has four neurons, which are designed as complex/real-valued for comparison. Intensity detection is performed to the first layer. The second layer is a real-valued linear mapping between the hidden and output nodes. In chip implementation, we scan both inputs $x_1$ and $x_2$ from −1 to 1 by a step of 0.1. The output distribution ($y_1$, $y_2$) measured on the chip are shown, from which the classification results can be interpreted by the manner of *Argmax*: if $y_1 \geq y_2$, the instance belongs to the class blue, otherwise it belongs to the class pink. **b** The subfigures from left to right are the decision boundaries of the Circle, achieved, respectively, by the simulated real and complex model, as well as the chip implementation of complex model. In experiment results, the black wires are our expected decision boundaries, while the white wires are the experimental ones. The classification accuracy is displayed at the lower right corner of each subfigure. The same results of the Spiral are shown in **c**.

rate of 0.005, a training period of 100 iterations and a batch size of 100. The activation functions used in the real model and the complex model is ReLU and ModReLU, respectively (see Supplementary Note 14). The hidden layer is implemented on ONC, while the input layer (784 × 4) and the output layer (4 × 10) are executed electrically. Theoretically, the input and output layers are implementable on our ONC by decomposing a large matrix into multiple small matrices. Considering the practical workload and the same principles, we only focus on the proof-of-principle implementation of the hidden layer. Figure 6b shows the training process of complex-valued and real-valued models with same dimensions. The complex-valued model achieves a training accuracy of 93.1% and a testing accuracy of 90.5%, while the real-valued model obtains a training accuracy of only 84.3% and a testing accuracy of 82.0%. Their confusion matrices on testing

instances are shown in Fig. 6c, d. The complex-valued neural network significantly outperforms its real-valued counterpart. In addition, faster convergence is observed in the complex-valued model during the training process.

Part of the reason for the ~10% increase in the accuracy of the complex model is that it receives more information (both magnitude and phase) from the previous layer. Here, ablation studies are carried out to verify the contribution of the complex-valued weight matrix itself to the increase in accuracy, as well as to illustrate the roles of the encoding and detection methods in optical realization, by the following implementations on our ONC: (a) completely complex: both magnitude and phase are encoded and detected; (b) real encoding: only magnitude is encoded, but both magnitude and phase are detected (by coherent detection); (c) real detection: both magnitude and phase are

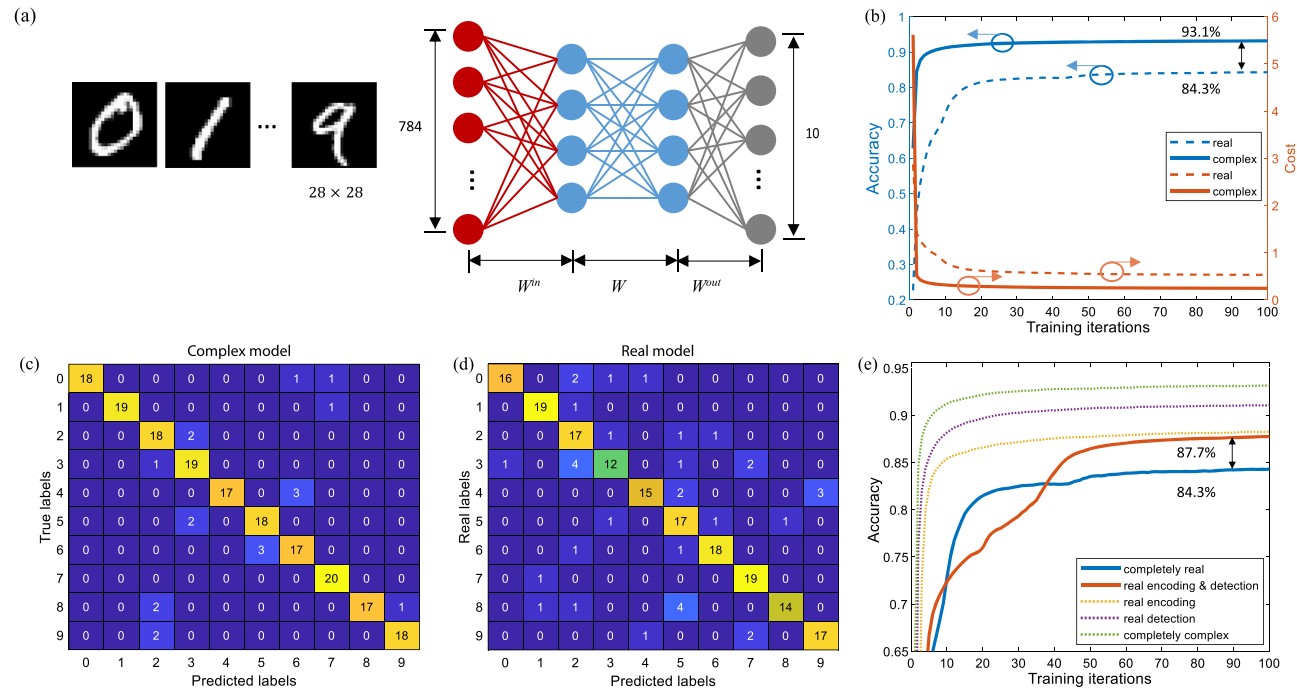

**Fig. 6 Handwriting recognition with a complex-valued multilayer perceptron. a** The network consists of an input layer $W^{in}$, a hidden layer $W$ and an output layer $W^{out}$. All 10 digits are included in our experiment. The input image sized as $28 \times 28$ is stretched into a $784 \times 1$ vector. The output layer maps the four hidden outputs to 10 classes. **b** The performance comparison of complex-valued and real-valued network implemented on the same chip. The blue and orange curves represent the accuracy and the cost of training, respectively. The solid line represents the complex-valued algorithm while the dashed line represents the real-valued algorithm. The training accuracy of complex-valued and real-valued models are 93.1% and 84.3%, respectively. In addition, a faster convergence is observed in complex model. **c** The confusion matrix by the chip-implemented complex model, when evaluating on testing set. Each column of the matrix represents the instances in a predicted label while each row represents the instances in a true label. The diagonal elements represent the number of instances that are correctly predicted. The chip testing accuracy is 90.5%. **d** The confusion matrix by the real model, showing a testing accuracy of 82.0%. **e** Sceneries are investigated, where complex-valued weights are implemented on chip, but input encoding and output detection are either complex-valued or real-valued. Their training curves are shown. Notably, even when we restrict both the encoding and detection to be real-valued, the complex-valued model exhibits a superior performance (87.7%) over its real-valued counterpart (84.3%).

**Table 1 Performance of complex-valued neural networks under different encoding and detection methods.**

| Size of hidden layer | Evaluation set | Completely complex (%) | Real encoding (%) | Real detection (%) | Real encoding & real detection (%) | Completely real (%) |
|---|---|---|---|---|---|---|
| $N = 4$ | Training | 93.1 | 88.3 | 91.1 | 87.7 | 84.3 |
| | Testing | 90.5 | 87.0 | 88.5 | 86.5 | 82.0 |
| $N = 8$ | Training | 96.0 | 93.1 | 96.9 | 93.6 | 92.3 |
| | Testing | 93.5 | 91.0 | 93.0 | 91.5 | 91.0 |

encoded, but only the magnitude is detected (by intensity detection); (d) real encoding and real detection: only magnitude is encoded and detected, and (e) completely real: real weight matrix, real encoding and real detection. In either of the first four scenarios, the hidden weight matrix $W$ is parameterized by complex values. The intensity and coherent detection respectively correspond to the activation function $M(z) = ||z||$, and $\text{ModReLU}(z) = \text{ReLU}(||z|| + b)e^{i\theta_z}$. The confusion matrices under scenarios (b), (c) and (e) are shown in Supplementary Fig. 17. As shown in the training curves in Fig. 6e, the complex-valued model, even when both encoding and detection are implemented with real values, outperforms the real-valued model (86.5% vs. 82.0%). In other words, the complex-valued model outperforms its real-valued counterpart regardless of the encoding and detection methods. In addition, the complex encoding achieves better performance than the complex detection, indicating that enlarged input information has a greater contribution to improving the network performance, while the compromise on

detection method would have a relatively marginal impact. We also demonstrate that under comparable capacity, a complex model achieves higher accuracy. Here, the capacity is defined as the number of effective real-valued parameters contained in a network model, which is a function of the neuron number $N$. The performance of complex models with $N = 4$ and $N = 8$ is shown in Table 1. A $4 \times 4$ complex model (whose capacity is 32) beats an $8 \times 8$ real model (whose capacity is 64) in terms of accuracy (93.1% vs. 92.3%). To obtain comparable performance, the complex-valued model requires a smaller chip size and thereby requires fewer free components (12 PSs on a 4-mode complex-valued chip vs. 56 PSs on an 8-mode real-valued chip).

We compare the costs of optically implementing the complex and real neural network from three aspects: the input encoding, the weight multiplication and the detection method. Firstly, real encoding has the same cost as complex encoding, because real inputs to the optical chip can be regarded as complex inputs with their phases limited to 0 or $\pi$, which also requires the same

control over the relative phases between different paths. Therefore, complex encoding is more informative (encoding information on both magnitude and phase, and correspondingly doubling the hidden input), while occupying the same number of chip components. Secondly, the implementation of both real and complex weight matrices requires the modulation of all internal and external PSs. The only additional cost of the complex model comes from coherent detection, which requires twice the amount of measurements required for intensity detection. However, even if we only perform intensity detection (Table 1), the complex-valued model maintains accuracy of 88.5%, which is still significantly higher than the real-valued model (82.0%).

The scale of most practical neural networks on conventional electronic computers is difficult to achieve with current optical circuits, although it is potentially achievable by the further development of the industry foundary[59,60]. Our work highlights that complex-valued optical neural networks, although at a small scale, can achieve performance comparable to larger-scale real-valued implementations. In addition, our complex-valued implementations enhance the learning capability of optical networks without increasing the complexity of the hardware, thereby helping to alleviate the problems imposed by the current limitations of manufacturing large-scale optical chips.

## Discussion

Complex values in neural networks host a number of performance advantages but were burdened by the heavy computational costs of complex multiplication in conventional computers. Here, we have demonstrated the implementation of genuine complex-valued neural networks on a single ONC, where complex multiplication can be realized passively by optical interference. The resulting ONCs have significant performance advantages over real-valued counterparts in a range of tasks at both the single-neuron and the network level. Notably, a single complex-valued neuron is able to solve certain nonlinear tasks that cannot be done by its real-valued counterpart. Moreover, complex-valued networks on our ONCs demonstrate marked improvements in classification of nonlinear datasets and handwriting recognition tasks. The advantages are briefly concluded as: (a) It offers doubled number of trainable free parameters, using the same physical chip as real-valued networks. (b) It is capable of classifying nonlinear patterns with simple architectures, e.g., fewer layers and neurons, as well as achievable activation function ($M(z) = ||z||$ by intensity detection). Thus, we have illustrated the potential of complex-valued optical neural networks to feature versatile representations, easy optimization and rapid learning. Meanwhile, the small chip size, low cost, high computational speed and low power consumption make it practical to implement large-scale optical deep learning algorithms on our ONC.

Our ONC also provides a natural pathway towards the near-term quantum computation. Notably, our perceptrons here are realized by networks of optical interferometers. Such networks—when coupled with non-classical light—can enable sampling tasks that are classically intractable. Indeed, there has been a number of recent proposals in generalizing neural networks to the quantum domain[61], such as the quantum optical neural network which utilizes high dimensionality of Boson sampling distributions[62]. Our platform, with the incorporation of non-classical light sources (e.g., single photon Fock states) and photon number resolving detectors, thus provides a promising avenue for their realization. Our platform can also be used to demonstrate some specific algorithms, such as quantum variational autoencoder[63] and quantum generative adversarial networks[64].

We can extend our ONC into a fully fledged multilayer neural network by cascading the optical circuits. Thus our proposal

satisfy all the criteria of cascadable photonic neural networks, including isomorphism, physical cascadability, gain cascadability, and noise cascadability[22]. The isomorphism and physical cascadability are inherently satisfied since each neuron has a hardware counterpart on the ONC, and the output is in the same physical format as the input. Our multilayer proposal is assisted by an electrical interface that matches the gain of input and output with a good gain cascadability, which is distinct from all-optical configurations that demand high-gain optical-to-optical nonlinearity. However, because our neuron variables are represented by light waves, the noise is inevitably susceptible to many factors, including imprecise phase, photodetection noise, coupling drift, and thermal crosstalk. To achieve a good noise cascadability, besides carefully controlling the aforementioned factors, we can take the noise into account during training procedure and compensate the noise using validated strategies[65].

## Methods

**Experimental set-up**. The light source was a 1550-nm laser with 12 dBm power from a Santec TSL-510 tunable laser. A polarization controller was applied to maximize the coupling of the light source to the ONC. A Peltier controlled by Thorlabs TED200C was used to assist heat dissipation, stabilize the temperature of the chip and reduce the heat fluctuations caused by ambient temperature and the heat crosstalk within the chip. The data acquisition module included a gainable TIA and an Analogue-to-Digital convertor NI-9215 with a resolution of 16 bit. The performing circuit which provided the electrical power to PSs had a 16-bit output precision.

**Chip characterization**. The $I$–$V$ characteristics of each heater was calibrated. The relationship between electrical power and current were fitted by a non-resistive model $P(I) = p_1 I^3 + p_2 I^2 + p_3 I + p_4$. The characterization of each PS was done by varying the applied current while measuring the optical power at the output port. The collected measurement data were fitted with $y = -a \cdot \cos(b \cdot (P + c)) + d$, where $y$ was the optical power, $d$ was a constant background, $a$ was the maximum magnitude of the signal, $b$ and $c$ were coefficients depicting the relationship between the phase and the electrical power $P$ computed by the non-resistive model. An average $R$-square value of 0.99 was achieved with the fittings, which indicated that the model adequately reproduced the data observed from the measurements. The average visibility was 99.85%.

**Coherent detection**. The switching between the intensity and coherent detection is assisted by the rightmost column of MZIs with adjustable interference state (grey-coloured in Fig. 1b). For intensity detection, the MZI was configured with $\theta = \pi$. The electrical field of each signal light was converted to photocurrent by the photodiode placed at its end. For coherent detection, the PS $\theta$ was set to $\pi/2$ for maximum interference between the signal and reference light. By connecting the two photodiodes at both output ports of the MZI in a balanced way, the subtracted output current was $I_I \propto 2A_s A_l \cos\phi_s$, where $A_s$ and $A_l$ were the magnitudes of the signal and the reference light. Similarly, by adding a phase shift of $\pi/2$ to the reference light, the output current was $I_Q \propto 2A_s A_l \sin\phi_s$. The $\phi_s$ was then determined from the ratio of $I_I$ and $I_Q$, which also eliminates the physical noises from the optical components. Instead of coupling the chip to an off-chip 90° optical hybrid that was conventionally used for coherent detection, our on-chip coherent detection avoided the phase fluctuating caused by fibre coupling and improved the stability and reliability of coherent detection with simplified experimental setup.

## Data availability

The data that support the findings of this study are available from the corresponding authors on reasonable request.

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

## Acknowledgements

This work was supported by the Singapore Ministry of Education (MOE) Tier 3 grant (MOE2017-T3-1-001), the Singapore National Research Foundation (NRF) National

Natural Science Foundation of China (NSFC) joint grant (NRF2017NRF-NSFC002-014) and the Singapore National Research Foundation under the Competitive Research Programme (NRF-CRP13-2014-01), the Quantum Engineering Programme (QEP-SF3), the NRF fellowship (NRF-NRFF2016-02) and the NRF-ANR joint programme (NRF2017-NRF-ANR004) VanQuTe.

## Author contributions

H.Z., X.D.J., L.C.K. and A.Q.L. jointly conceived the idea. H.Z. and H.C. designed the chip and built the experimental setup. H.C., G.Q.L., B.D., X.S.L. and D.L.K. fabricated the silicon photonic chip. H.Z., H.C., F.K.M. performed the experiments. S.P., R.S., and A.L. assisted the set-up and experiment. M.G., J.T., Y.Z., M.H.Y., X.D.J and L.C.K. assisted with the theory. All authors contributed to the discussion of experimental results. X.D.J., L.C.K. and A.Q.L. supervised and coordinated all the work. H.Z., M.G., J.T., Y.Z., Y.Z.S., X.D.J., L.C.K. and A.Q.L. wrote the manuscript with contributions from all co-authors.

## Competing interests

The authors declare no competing interests.
