## [Peer Review File · Nature Communications]

Reviewers' Comments:

Reviewer #1:

Remarks to the Author:

This work demonstrates some key functions, i.e. matrix multiplications, (but only a subset of the functions) of a complex-valued optical neural network (PNN) using an MZI mesh network fabricated on silicon photonic platforms. This work also compares the advantages of complex-valued PNN over real-valued PNN using some benchmark tasks. However, some key information is not disclosed or not appropriately presented. There are also several important inconsistent descriptions in the manuscript. These make it challenging to evaluate the true contribution and creditability of this work. Details are shown below.

Introduction

1. The work adopts an MZI mesh network similar to Ref.[32].

There are several work using coherent detection in PNNs, such as Hamerly, Ryan, et al. "Large-Scale Optical Neural-Network Accelerators based on Coherent Detection." CLEO:

QELS_Fundamental Science. Optical Society of America, 2019, and Ishihara, Tohru, et al. "An Optical Neural Network Architecture based on Highly Parallelized WDM-Multiplier-Accumulator." 2019 IEEE/ACM Workshop on Photonics-Optics Technology Oriented Networking, Information and Computing Systems (PHOTONICS). IEEE, 2019.

The authors should give sufficient credits to these work.

2. The authors claim that "conventional electronic computing platforms exhibit significant slowdown when executing algorithms using complex-valued operations because complex numbers have to be represented by two real number". Here, authors should clearly state what electronic platforms they refer to. For example, this claim does not apply to analog electronic platforms. Again, I would suggest the author acknowledge the efforts in analog electronic neural networks.

3. The authors claim that "previous complications of complex-valued networks – cumbersome arithmetic on complex numbers – are alleviated by directly passively realizing such operations through optical interference with no computational overhead." First, it is not clear what computational overhead the authors have in mind here. Second, it should be aware that PNN has nonnegligible computation overhead, such as the interface between optics and signals involving heavy OEO conversions and ADCs.

4. The authors claim that PNN has "potential implementations of high dimensional quantum neural networks." Again, I think this claim is a bit aggressive without solid ground provided.

Design and Fabrication

1. Generally, the operation and implementation of the photonic circuit are not clearly presented. Some key information is not disclosed. Some key descriptions are not precise.

Here are some examples:

1. How is the local oscillator generated? From the circuit, it seems that the local oscillator has gone through a chain of phase shifters. How can you make sure the local oscillator is not modulated by those phase shifters?

2. What detection scheme is used here? At someplace, the authors mentioned intensity detection. But at the end of this section, it says coherent detection. If it is coherent detection, only a single photodetector is shown in Fig.2. Textbooks tell us coherent detection needs a coherent detector consists of four photodiodes together with a 90-degree optical hybrids.

3. It is very unclear how the nonlinear activation function is implemented. The authors said: The activation function adopted is detection-based nonlinear activation which can be realized by either inline monitors together with additional set of interferometers at the end or simply remodulate the inputs according to the detection results of the output signals.

What is inline monitor and how is it fabricated? What is the function of inline monitor here? What do additional set of interferometers do here? I think the authors did not make efforts in presenting their ideas.

4. $U(N)$ should be an $N \times N$ matrix. However, from the expression of $U(N)$ vs. T (line 1, page 7), the resulting $U(N)$ is a 2×2 matrix because T is a 2×2 matrix.

2. Some concerns about the scheme:

1. The precision of phase shift implementation in complex-valued PNNs.

a. In the Supplementary, according to the values of the phases on each MZI, the phase needs to be implemented with an accuracy of 0.01%. However, the authors also claim that the photodetector noise would degrade the accuracy to 0.02. What is the exact implementation accuracy of the whole scheme? I guess the accuracy should be limited by the lower bound which is 0.02.

b. The authors characterize the thermal crosstalk between the heaters by heating up one heater while measuring its influence on the other heater. However, there are many heaters on the chip, thus such crosstalk will accumulate. In addition, the authors only consider the crosstalk caused by heat leaking. I notice that many heaters share the same ground, in which parasitic resistors would exist and have a large impact on the thermal crosstalk.

c. How would the inaccurate phase shift implementation impact the accuracy of artificial neural network tasks? The authors achieved an accuracy of 99.3% on chip which is very impressive, while Ref. [32] only achieve an accuracy of 76% (although on a different task). I am wondering how accurate the phase shift implementation should be to achieve such a high accuracy at 99.3%.

2. Cascading the PNN

a. Cascading PNN cannot be simply described as modulating the input of the next layer using the previous layer as the authors did in their manuscript. De Lima, Thomas Ferreira, et al. "Machine learning with neuromorphic photonics." *Journal of Lightwave Technology* 37.5 (2019): 1515-1534. provides the criterion on whether a PNN is cascading or not.

b. What does feedback-clocked strategy mean here? How does the clock work with this analog system?

Results:

1. The procedures for implementing the neural network tasks are not clear. The authors should clearly state:

1. In the general description part i.e. the first paragraph of Results:

a. How is complex-valued bias implemented on-chip?

b. What is the activation function used, how is it implemented especially on the complex values?

2. For each task,

a. How are the inputs mapped to the optical domain?

b. What algorithm is used for training?

c. What is the cost function and objective value for each task, especially considering the output of the neural network is a complex value.

d. What and how is the activation function used for each task.

2. Some minor questions:

1. The results of XOR are not shown

2. When you achieve 99.3% for Iris task, how many samples are evaluated?

3. "Inner phase shifter" and "outer phase shifter" are not defined.

Finally, there are some mistakes in the Supplementary

1. The carrier-injection modulator is challenging to operate at 10 Gbit/s. I guess the authors want to say carrier-depletion modulation.
2. Cross-section of the waveguide figure: the notation of the heater is not correct. It should be 3 μm instead of 100 μm . "heater" should be on top of the yellow region instead of the grey region.

Reviewer #2:

Remarks to the Author:

1. The idea of computing with complex neural networks is not novel and has been investigated in the recent past in the work of Prucnal et. al. and in various reservoir computing implementations. The related increase in computational power related to real-valued approaches is also a well documented fact. The idea of training complex weights is also not new and for example was studied (albeit in simulation) in the work of Martin Fiers et. al. : Nanophotonic Reservoir Computing With Photonic Crystal Cavities to Generate Periodic Patterns and a number of other recent works (see also 10.1109/TNNLS.2018.2874571 that presents two complex-valued algorithms to training complex-valued Optical Neural Networks). Their implementation also borrows heavily from recent work on MAC units based on MZIs. The main contributions of the work seem to be a hardware implementation tested in various application scenarios and making a more detailed comparison of complex-valued neural networks with their real-valued counterparts. Specifically, the main claim of " We tackle these issues by proposing and experimentally realizing an optical neural chip (ONC) that executes complex-valued arithmetic ..." is exactly how other coherent light-based Optical NN implementations operate. Putting all this in perspective, it is not clear to me that this work represents novel contributions at the level expected by the audience of Nature Communications on the subject of Neuromorphic Photonics. This however does not, of course, downplay the authors' commendable engineering effort.
2. The authors have also not addressed the issue of how the loading phase of the data (from high speed memory?) to be trained onto their optical neural network could impact the power efficiency and speed of operation, especially as the number of MZI units and bandwidth increases.
3. The scaling approaches outlined by the authors towards multi-layer Optical NNs seem to introduce extensive energy efficiency challenges and operational complexity that may hit the practical limits. For example the strategy for faster operation based on free carrier modulation effects would be quite challenging to implement. For example 10GHz (or higher) signals require expensive RF I/O equipment and high speed interfaces (for example each phase shifter in each modulator would require its own 10GHz RF source signal). The other components of the system would also become more challenging to deal with e.g. multiple high speed ADCS, TIAs. Are there potential was around this?
4. The work needs very thorough proof-reading as there are numerous grammatical and other errors that some times make the manuscript challenging to follow.
5. In response to one of the earlier reviewers' questions about power efficiency, the authors claim that "once trained and configured, the system does not consume any power" (Round 1_reviewer 3 comment 4). Isn't this only the case if the weights (the modulator phases) are non-volatile. Don't you still consume power to maintain the specified phase configuration? And how much?
6. Regarding the example on hand writing recognition, do the authors envision a feasible way to integrate the input weight matrix (784 x 4) on the same chip as their Optical Neural Network? Are there any significant computational benefits to doing this?
7. I find the lower most schematic in Figure 1 confusing to interpret as it does not directly map to figure above it. It might help to make it a figure of its own and caption it accordingly.

Reviewer #3:

Remarks to the Author:

The authors reported a theoretical proposal and experimental proof-of-concept of an integrated-complex value optical matrix multiplier and applied the chip to solve three different machine learning tasks (the rest of the algorithms in those tasks were implemented electronically).

The paper is a natural extension of Ref. 32 and is well written. The topic of complex optical neural networks is quite intriguing and makes a lot of sense in the roadmap of optical computing. Because coherent optical signals are inherently complex numbers. The author has used a similar thermal phase shifters to do the data and weight encoding and implemented coherent detections to detect complex output. However, after reading through the whole paper, I do have some major questions and need the authors to address them in order for me to recommend publication of this paper on Nat. Comm.

1. Experimental demonstration:

It would be good if the author can clarify what was done optically and what was done electronically in their experiment section. To my understanding, the W_{in} and W_{out} are both done in the electronic domain, while only the middle 4×4 matrix was done optically. So how many weights are in W_{in} and W_{out} ? If W_{in} and W_{out} are a lot larger than 4×4 , then it is hard to argue the computation was done optically. Also, in theory, one can always break a bigger matrix (including W_{in} and W_{out}) into a smaller matrix and implement them optically, why didn't the author do that?

2. Algorithm scale-up:

It is interesting to see that the complex neural networks idea starts working for the benchmarks discussed in the paper. However, all these are still considered toy tasks, and it is not very convincing to me that complex NN really works better than real-valued NN. There are better "real value" based neural networks that perform much better than the results referenced in the paper (for example, CNN can achieve >99.8% accuracy on MNIST). Therefore, it would be more interesting to see if the authors can take a state of the arts real-valued model, and improve it with complex numbers. Moreover, it would be good to see if complex NN can really work on some more realistic datasets, for example, Imagenet or CIFAR tasks. Experimental demonstration is not needed for those bigger datasets, just numerical result should be sufficient.

In conclusion, I really like the concept raised in this paper and think it is an interesting direction and would recommend the publication of this paper if authors can satisfactorily answer the questions above.

Manuscript ID:	Nature Communications manuscript NCOMMS-20-14526
Paper title:	An Optical Neural Chip for Implementing Complex-valued Neural Network
Authors:	H. Zhang, M. Gu, X. D. Jiang, J. Thompson, H. Cai, S. Paesani, R. Santagati, A. Laing, Y. Zhang, M. H. Yung, F. K. Muhammad, G. Q. Lo, X. S. Luo, B. Dong, D. L. Kwong, L. C. Kwek, and A. Q. Liu

Reply to Reviewer 1

We are grateful to the Reviewers for their constructive comments, and their recognition of our contribution of on-chip integration of key components for matrix multiplications for implementation of complex-valued optical neural networks (ONN). The Reviewer still had several concerns about detailed design and experimental realization of the scheme. We are happy to address all such concerns in lines below. We thank the Reviewer for the opportunity and suggestions to improve our manuscript.

Comment 1: *The work adopts an MZI mesh network similar to Ref.[32]. There are several work using coherent detection in PNNs, such as Hamerly, Ryan, et al. "Large-Scale Optical Neural Network Accelerators based on Coherent Detection." CLEO: QELS_Fundamental Science. Optical Society of America, 2019, and Ishihara, Tohru, et al. "An Optical Neural Network Architecture based on Highly Parallelized WDM-Multiplier-Accumulator." 2019 IEEE/ACM Workshop on Photonics-Optics Technology Oriented Networking, Information and Computing Systems (PHOTONICS). IEEE, 2019. The authors should give sufficient credits to these works.*

Answer: As suggested, we have included a discussion of both in our manuscript (see below). The relation of these two papers to our work is indeed a relevant and interesting point. However, we do note that there are several salient differences between these schemes and our implementation.

We give credits to those two works by discussing their contributions and clarifying their difference with our work in Line 9 on Page 4 as

"High parallelized optical neural network accelerator based on photoelectric multiplication was reported^{31, 32}. This acceleration is designed for real-valued networks because the optical signals had already been converted to photocurrents by the homodyne detectors before reaching the accumulator. "

And in Supplementary information 1: Related works in Line 10 on page 2 as

"Another optical neural network accelerator based on photoelectric multiplication was reported². Unlike the previously mentioned approaches, this scheme encoded both weights and inputs optically, and the matrix multiplication was realized by combining the

input and weight signals for balanced homodyne detection. Although involving coherent detection, the network architecture applicable to the accelerator was still real-valued, because the optical signals had already been converted to photocurrents by the detectors before reaching the accumulator ($\sum_{j=1}^n |A_j| |x_j|$ vs. $\sum_{j=1}^n |A_j| |x_j| e^{i(\phi_A + \phi_x)}$, where $A_j = |A_j| e^{i\phi_A}$ and $x_j = |x_j| e^{i\phi_x}$). Motivated by the parallelization capability of this scheme, an architecture using wavelength division multiplexing for vector-matrix multiplication was proposed³. The architecture fully exploited optical parallelization and showed promise for low-power and low-latency optical neural networks. However, their goal was distinct from ours in seeking to realize parallel computing while we are actually designing a complex neural network.”

Comment 2: *The authors claim that “conventional electronic computing platforms exhibit significant slowdown when executing algorithms using complex-valued operations because complex numbers have to be represented by two real number”. Here, authors should clearly state what electronic platforms they refer to. For example, this claim does not apply to analog electronic platforms. Again, I would suggest the author acknowledge the efforts in analog electronic neural networks.*

Answer: When writing this, we did indeed have digital electronic platforms in mind. The referee is also on point on acknowledging analog electronic devices. In response we have

- Included a qualifying to make it clear are referring to digital computing platforms. Line 15 on Page 3 now reads “Conventional digital electronic computing platforms exhibit significant slowdown when executing algorithms using complex-valued operations because complex numbers have to be represented by two real numbers^{9, 15}.”
- Acknowledge the efforts in analog electronic neural networks in manuscript in Line 15 on Page 4 as “Besides optical computing platforms, analog electronic devices, as opposed to the digital electronic devices, have successfully demonstrated multi-layer perceptron^{40, 41} and convolutional neural networks⁴². Complex-valued neural networks on analog electronic devices have already been explored in some pioneer works⁴³⁻⁴⁵.”

Comment 3: *The authors claim that “previous complications of complex-valued networks – cumbersome arithmetic on complex numbers – are alleviated by directly passively realizing such operations through optical interference with no computational overhead.” First, it is not clear what computational overhead the authors have in mind here. Second, it should be aware that PNN has nonnegligible computation overhead, such as the interface between optics and signals involving heavy OEO conversions and ADCs.*

Answer: First we would like to explain what the computational overhead refers to. The computational overhead is the additional multiplication-accumulation operations (MACs)

required when training a complex-valued neural network using digital electronic devices. For instance, the multiplication of two complex numbers, $(a + ib) \cdot (x + iy) = (ax - by) + i(bx + ay)$, requires 4 multiplication and 2 addition operations. A detailed analysis is available in the book “Complex-Valued Neural Networks with Multi-Valued Neurons” by Aizenberg, et.al, and “Representation of complex-valued neural networks: a real-valued approach” by Yadav, A, et.al. In contrast, optical neural chip carries coherent optical signals which are inherently complex numbers. As a result, complex-valued neural network can be readily implemented with our optical neural chip without added computational cost.

We have explained the cumbersome implementation of complex-valued neural networks by digital electronic computing platform in the manuscript as

- *“conventional digital electronic computing platforms exhibit significant slowdown when executing algorithms using complex-valued operations because complex numbers have to be represented by two real numbers^{9, 15}, which increases the number of multiply-accumulate operations – the most frequently used while computationally expensive component of the neural network algorithms^{16, 17}.” in Line 15 on Page 3.*
- *“Meanwhile, previous complications of complex-valued networks – cumbersome arithmetic on complex numbers – are alleviated by directly passively realizing such operations through optical interference.” in Line 13 on Page 5.*

We hope to emphasize the advantages of using integrated optical neural chip for implementing complex-valued neural networks, and at the same time not to overlook the fact that optical neural networks have non-negligible computation costs at the electrical-optical interface and ADCs, as well as the standing costs of the optical neural chip itself.

Comment 4: *The authors claim that PNN has “potential implementations of high dimensional quantum neural networks.” Again, I think this claim is a bit aggressive without solid ground provided.*

Answer: As suggested by the Reviewer, we summarize the aforementioned backgrounds and recent developments of high-dimensional quantum neural networks in Discussion as

“Indeed, there has been a number of recent proposals in generalising neural networks to the quantum domain⁶⁶, such as the quantum optical neural network which utilizes high dimensionality of boson sampling distributions⁶⁷. Our platform, with the incorporation of non-classical light sources (e.g. single photon Fock states) and photon number resolving detectors, thus provides a promising avenue for their realization. Our platform can also be used to demonstrate some specific algorithms, such as quantum variational autoencoder⁶⁸ and quantum generative adversarial networks⁶⁹.” in Line 4 on Page 17.

Comment 5: *How is the local oscillator generated? From the circuit, it seems that the local oscillator has gone through a chain of phase shifters. How can you make sure the local oscillator is not modulated by those phase shifters?*

Answer: The local oscillator, as well as the signal lights, are generated on-chip from a single coherent light source connected to the chip, instead of being generated off-chip and then guided into the waveguides. As suggested by the Reviewer, we clarify the process of generating the local oscillator, as well as the necessity of integration of the input division and modulation in the revised manuscript. Stating on Line 7 on Page 8:

“The local oscillator, as well as signal lights, are generated on chip from a single coherent light source and is modulated by the same chain of phase shifters. The coherent optical neural chip that implements either complex-valued or real-valued neural networks requires stringent control over the phase of the light signals. The integration of the light division and modulation effectively avoids the possible phase fluctuations which take place when coupling external light signals to the chip.”

Comment 6: *What detection scheme is used here? At someplace, the authors mentioned intensity detection. But at the end of this section, it says coherent detection. If it is coherent detection, only a single photodetector is shown in Fig.2. Textbooks tell us coherent detection needs a coherent detector consists of four photodiodes together with a 90-degree optical hybrids.*

Answer: Our answers to the three questions raised by the Reviewer are:

- Both intensity detection and coherent detection are used in our work. The choice of detection method is determined by the nonlinear activation function applied. Intensity detection is naturally applied for $M(z) = \|z\|$, whereas coherent detection is applied for $\text{ModReLU}(z) = \text{ReLU}(\|z\| + b)e^{i\theta z}$.
- Instead of using a 90-degree hybrid, we perform on-chip coherent detection to simplify the experiment setup and improve phase stabilization. Each output port of the circuit is connected to a photodiode, but not all are shown due to limited space. Every two photodiodes are connected in a balanced way, which means that the two photodiodes are reverse biased by connecting the positive port of one to the negative port of the other, for homodyne detection. The coherent detection is done with two separate measurements, one with a $\pi/2$ phase applied to the reference light and the other without, as compared to the conventional 90-degree configuration which uses 4 photodiodes and does the coherent detection with a single measurement. The details of our coherent detection strategy are provided in the Supplementary information 7 and Fig. S8.

As suggested by the Reviewer, we provide a more detailed description of the detection scheme using our integrated photonic chip in the revised manuscript as

“Our integrated chip is capable of performing both intensity detection and coherent detection. The choice of detection method is determined by the nonlinear activation function applied. Intensity detection is naturally applied for the activation function $M(z) = \|z\|$, whereas coherent detection is applied for the activation function $\text{ModReLU}(z) = \text{ReLU}(\|z\| + b)e^{i\theta z}$.” In Line 8 on Page 8.

And the experimental details in Methods: Coherent detection as

“The switching between the intensity and coherent detection is assisted by the rightmost column of MZIs with adjustable interference state (grey-coloured in Fig.1b). For intensity detection, the MZI was configured with $\theta = \pi$. The electrical field of each signal light is converted to photocurrent by the photodiode placed at its end. For coherent detection, the phase shifter θ was set to $\pi/2$ for maximum interference between the signal and the reference light. By connecting the two photodiodes at both output ports of the MZI in a balanced way, the subtracted output current is $I_I \propto 2A_s A_l \cos\phi_s$, where A_s and A_l are the magnitudes of the signal and the reference light. Similarly, by adding a phase shift of $\pi/2$ to the reference light, the output current is $I_Q \propto 2A_s A_l \sin\phi_s$. Therefore, the ϕ_s can be determined. Instead of coupling the chip to an off-chip 90-degree optical hybrid that is conventionally used for coherent detection, our on-chip coherent detection avoids the phase fluctuating caused by fibre coupling and improves the stability and reliability of coherent detection with simplified experimental setup.” In Line 21 on Page 18.

Comment 7: *The authors said: The activation function adopted is detection-based nonlinear activation which can be realized by either inline monitors together with additional set of interferometers at the end or simply remodulate the inputs according to the detection results of the output signals. What is inline monitor and how is it fabricated? What is the function of inline monitor here? What do additional set of interferometers do here? I think the authors did not make efforts in presenting their ideas.*

Answer: Our Our answers to the three questions raised by the Reviewer are:

- Inline monitors are photodiodes fabricated on one ports of the connecting MZIs (refer to Fig. S10d).
- The functionality of inline monitor is to perform detection. Activation function will be applied electrically to the detection results.
- The connected interferometers are used to modulate the signal light according to the electrically computed results of the applied activation function.

As suggested by the Reviewer, we describe how our detection-based activation functions are applied in multi-layer networks in Supplementary information 9. Stating in Line 18 on Page 15:

“Our nonlinearity functions are achieved by converting optical signal to electronic signal first, applying pointwise activations and converting the electrical signal back to optical

signal. The choice of the nonlinear activation function is dependent on the measurement methods. Intensity-based activations such as $M(z) = \|z\|$ require for intensity detection. Other functions such as the hyperbolic tangent and Rectifier linear unit (ReLU) variations require phase-sensitive detection. The multi-layer neural network structure is assisted by an electrical interface^{17, 38-40}. A column of reconfigurable MZIs at the end of each path are used to connect the layers. The functionality of the MZIs switches between detection and modulation. An inline monitor (photodiode or grating structure) is placed at the cross port of each MZI (see Fig. S10d). The MZIs are originally set to a state that the signal lights are transmitted entirely from the cross port for detection. Activation function is applied to the acquired electrical signals. The MZIs are then reconfigured to modulate the signal lights according to the results after the activation function is applied. Another commonly used method is to draw a small portion of the signal light from the current layer for detection and modulate the remaining signal light that is left to enter the next layer⁶⁵⁻⁶⁷."

Fig. S10d | The connection between two optical layers. A column of reconfigurable MZIs are used to connect two optical layers. By reconfiguring the MZI with $\theta = 0$, the signal light is transmitted entirely from the cross port and detected by the inline monitor. Being aware of the output from the n^{th} hidden layer, we apply activation function electrically and reconfigure the connected MZI to generate new input for the $(n+1)^{\text{th}}$ hidden layer. This is the procedure of a detection-based implementation of activation function.

Comment 8: $U(N)$ should be an $N \times N$ matrix. However, from the expression of $U(N)$ vs. T (line 1, page 7), the resulting $U(N)$ is a 2×2 matrix because T is a 2×2 matrix.

Answer: The dimensionality of $U(N)$ is $N \times N$ in terms of the N input modes. As suggested by the Reviewer, to be more accurate, we revise the relevant paragraphs as

"A N -mode network realizes the weight matrix multiplication by transforming the input states into an output according to $S_{out} = U(N)S_{in}$. $U(N)$ is a $N \times N$ unitary matrix represented as a product of rotation matrices $\{T_{pq}\}$ and a diagonal matrix D such that $U(N) = \prod_{p=2}^N \prod_{q=1}^{p-1} T_{pq} D$, where T_{pq} is defined as the N -dimensional identity matrix with the elements t_{pp} , t_{pq} , t_{qp} and t_{qq} replaced by

$$\begin{bmatrix} t_{pp} & t_{pq} \\ t_{qp} & t_{qq} \end{bmatrix} = ie^{i\frac{\theta}{2}} \begin{bmatrix} e^{i\phi} \sin \frac{\theta}{2} & e^{i\phi} \cos \frac{\theta}{2} \\ \cos \frac{\theta}{2} & -\sin \frac{\theta}{2} \end{bmatrix} \quad (1)$$

where θ is defined as the internal phase shifter between two beam splitters and ϕ as the external phase shifter at the end of the MZI.” In Line 15 on Page 7.

Comment 9: *The precision of phase shift implementation in complex-valued PNNs.*

a. In the Supplementary, according to the values of the phases on each MZI, the phase needs to be implemented with an accuracy of 0.01%. However, the authors also claim that the photodetector noise would degrade the accuracy to 0.02. What is the exact implementation accuracy of the whole scheme? I guess the accuracy should be limited by the lower bound which is 0.02.

Answer: The exact implementation of the whole scheme is 0.02 rad as degraded by photodetection. The 0.01% accuracy is not a requirement but rather a best-case scenario used in theoretical calculation.

Stating in Supplementary information 5 in Line 14 on Page 9 as

“Although we expect the implementation of phase shifts to be highly accurate, in actual experiments the accuracy of phase shifts deviates from the best-case scenario according to the physical components and signal distortion is unavoidable. We infer the resolution of phase detection from the photodetection resolution. The normalized photodetection noise is $\sigma_D \approx 0.6\%$ based on our observation. Therefore, the detection limit ($3\sigma_D$) of the photodetection is ~ 0.02 . The output intensity of an MZI with respect to its inner phase shifter is sinusoidal. Under the small-angle approximation of $\sin(\theta) = \theta$, the phase resolution $\Delta\theta = 0.02$ rad. Therefore, the limit of our setup in implementing small phase shift as inferred from the intensity is 0.02 rad.”

Comment 10: *The authors characterize the thermal crosstalk between the heaters by heating up one heater while measuring its influence on the other heater. However, there are many heaters on the chip, thus such crosstalk will accumulate. In addition, the authors only consider the crosstalk caused by heat leaking. I notice that many heaters share the same ground, in which parasitic resistors would exist and have a large impact on the thermal crosstalk.*

Answer: Our solutions to crosstalk includes:

- Trenches for mitigate the influence of adjacent heaters.
- An external cooling system for the heat dissipation, as described in Methods: Experimental set-up in Line 4 on Page 18: “A Peltier controlled by Thorlabs TED200C was used to stabilize the temperature of the chip and reduce the heat fluctuations caused by ambient temperature and the heat crosstalk within the chip.”

- For electrical crosstalk, we use a current driver instead of a voltage driver for the heaters. The content is added in Supplementary information 3 in Line 6 on Page 6

“Electrical crosstalk usually take place due to multiple phase shifters sharing a common ground. We investigate a simplified model where n phase shifters with effective resistance of R_1 are connected to the same ground, and the inevitable wire resistance is denoted as R_2 . The actual electrical power supplied to each phase shifter is $P_{1,act} = \frac{R_1 V_{cc}^2}{(R_1 + nR_2)^2}$, while we expect $P_{1,exp} = \frac{V_{cc}^2}{R_1}$. The parasitic resistance in the circuit $R_p = 2nR_2 + \frac{n^2 R_2^2}{R_1}$. Therefore, if the voltage drive is used, one must make sure that $R_2 \ll R_1$ or accurately calibrate R_2 . In contrast, current drivers avoid such problems by providing an electrical power $P_{1,act} = P_{1,exp} = I_{cc}^2 R_1$. Note that nI_{cc} is the current going through the wire resistor R_2 . To ensure that the current does not exceed the capacity of the circuit, our chip is designed in such a way that every four phase shifters share a common ground.”

The comparison between a voltage driver and a current driver.

Comment 11: How would the inaccurate phase shift implementation impact the accuracy of artificial neural network tasks? The authors achieved an accuracy of 99.3% on chip which is very impressive, while Ref. [32] only achieve an accuracy of 76% (although on a different task). I am wondering how accurate the phase shift implementation should be to achieve such a high accuracy at 99.3%.

Answer: First, we would like to clarify that the accuracy of 99.3% of a complex-valued layer is benchmarked against a real-valued neural network with an accuracy of 97.3%, and both values are obtained by numerical modelling to show the advantage of the algorithms without taking experimental errors into consideration. The chip testing results are shown in Fig. 4b, 4c and 4d with accuracies of 92.1%, 89.5% and 97.4%. We have revised the manuscript to make this point clearer as follows:

- Inclusion of the sentence, “*The accuracies of the three blind tests are 92.1%, 89.5% and 97.4%, respectively.*” In Line 4 on Page 13.
- Inclusion of the sentence “*The accuracy of a numerically simulated single complex-valued layer is benchmarked against its real-valued counterpart. The complex-valued layer achieves an accuracy of 99.3%, which outperforms the real-valued architecture that has an accuracy of 97.3%^{59, 60}, on the entire dataset with 150 samples. (see training curves in Fig. S13).*” In Line 6 on Page 13.

We would like to answer the Reviewer's question about our higher accuracy relative to Ref. [32] from three aspects:

- As the Reviewer also noticed, we implement different tasks of different complexities. Our Iris task has three species and two of which have overlaps, whereas theirs has four species, three of which are overlapped.
- Our implementation adopts a single layer, whereas theirs adopts four layers. The cumulative error would aggravate the effect on the final classification results, referring to B. Shi, et.al., “*Numerical simulation of an InP photonic integrated cross-connect for deep neural networks on chip*” (Applied Sciences, 2020).
- Ref. [32] actually expects a higher performance (90%) of their chip based on the noise level of each individual MZI, as stated in their discussions. They attributed the performance gap between 76.7% and 90% to the fact that their full ONN suffered from thermal crosstalk and proposed to compensate the crosstalk through additional calibration or by adding thermal isolation trenches. In our experiment, we adopt the isolation trenches as well as an external cooling system.

For a more reasonable comparison, we cite some works on optical neural networks that report a similar accuracy of the Iris classification task, such as the 97% accuracy reported by Ref. [59] by using a three-layer optical neural network, and the 91.6% accuracy by Ref. [60] by using a three-layer structure. Additionally in Supplementary information 12, we study on the *Iris* dataset about the impact of inaccurate phase shift. In particular, we applied different levels (standard deviation of 5%, 10%, 15%) of random noise are added onto the phase shift. As observed, 5% noise will cause a ~1% decrease in prediction accuracy. If the noise level exceeds 10%, the accuracy will greatly reduce. In multiple layers, cumulative error would aggravate the deviation of prediction accuracy in such a way that the prediction accuracy decreases as the number of layers of the photonic neural network increases.

Comment 12: *Cascading PNN cannot be simply described as modulating the input of the next layer using the previous layer as the authors did in their manuscript. De Lima, Thomas Ferreira, et al. "Machine learning with neuromorphic photonics." Journal of Lightwave Technology 37.5 (2019): 1515-1534. provides the criterion on whether a PNN is cascable or not.*

b. What does feedback-clocked strategy mean here? How does the clock work with this analog system?

Answer:

a. We carefully studied the reference provided by the Reviewer and summarized the criteria for cascability of optical neural networks. We explain our proposed way of cascading and evaluate if it meets these criteria as:

- *"Our proposed multi-layer neural network structure is assisted by an electrical interface. The outputs from the upstream layer undergo an optical-electrical (O-E) conversion. The activation function is applied electrically on the detection (coherent/intensity) results. The electrical signal is converted back to the optical signal as the input of the downstream layer. For a neural network has 5 layers with 8 neurons in each, we can use a chip that integrates 5 optical circuits, each of which has 8 input modes. An alternative approach is to use the same optical circuit to emulate a 5-layer network by iteratively storing its output, re-modulating the inputs, reconfiguring the weights, and repeating these procedures 5 times."* In Line 18 on Page 16 in Supplementary information 9.
- *"Our ONC can be easily extended into a fully-fledged multi-layer neural network by cascading the optical circuits. It can be configured to satisfy all criteria of a cascable photonic neural network including isomorphism, physical cascability, gain cascability, and noise cascability²². The isomorphism and physical cascability are inherently satisfied as each neuron has a hardware counterpart on the optical neural chip and the output is in the same physical format as the input. Our multi-layer proposal is assisted by an electrical interface that matches the gain of input and output with a good gain cascability, which is distinct from all-optical configurations that demand high-gain optical-to-optical nonlinearity. Because neuron variables are represented by light waves, the noise is susceptible to a number of factors including imprecise phase, photodetection noise, coupling drift, and thermal crosstalk. To achieve a good noise cascability, besides carefully controlling the aforementioned factors, we can take the noise into account during training and compensate the noise using validated strategies^{67,68}."* In Line 11 on Page 17 in Discussion.

b. The feedback strategy is describe as: First, an electrical processor feeds the optical chip with parameters for implementing the current layer. Next, the optical chip accomplishes the inference and sends results back to the electrical processor. Last, the electrical processor applies an activation function and re-modulates the chip to implement the

downstream layer. The above process can be iterated to implement a multilayer neural network.

Comment 13: *How is complex-valued bias implemented on-chip?*

Answer: We have described how we implement the complex-valued bias on chip as

“When implementing fundamental logic gates, the inputs to the neuron are real-valued, which could be considered complex values with phase being constrained to 0 or π . The bias is implemented with an additional constant input 1 weighted by a complex-valued weight b . Equation (2) is simplified to $\mathbf{y} = f(\mathbf{x} \cdot \mathbf{W})$, where the weight vector $\mathbf{W} = [w_1, w_2, \dots, w_n, b]^T \in \mathbb{C}^{(n+1) \times 1}$ and the input vector $\mathbf{x} = [x_1, x_2, \dots, x_n, 1] \in \mathbb{C}^{1 \times (n+1)}$.” In Line 12 on Page 10.

Comment 14: *What is the activation function used, how is it implemented especially on the complex values?*

Answer: Two kinds of activation function are applied and investigated in our paper. One is the intensity-detection-based $M(z) = \|z\|$, and the other is the coherent-detection-based $\text{ModReLU}(z) = \text{ReLU}(\|z\| + b)e^{i\theta z}$. The activation functions are applied electrically to the acquired output from the optical neural chip through intensity detection or coherent detection. More details are available in the response to comment 7.

Comment 15: *For each task,*

- a. How are the inputs mapped to the optical domain?*
- b. What algorithm is used for training?*
- c. What is the cost function and objective value for each task, especially considering the output of the neural network is a complex value.*
- d. What and how is the activation function used for each task.*

Answer: For these questions regarding the experimental details raised by the reviewer, we make a detailed explanation of each task and summarized these contents in Methods: Numerical simulation and implementation in Line 13 on Page 19.

“Logic gate realization:

The inputs of the binary logic gate in this tasks are four logic combinations, (0,0), (0,1), (1,0) and (1,1). As we treat the neuron bias as a result of an additional input, the inputs are correspondingly (0,0,1), (0,1,1), (1,0,1) and (1,1,1), where the first two bits are the logic inputs and the third bit is for bias. The inputs are encoded to the magnitude of three optical light waves while keeping them in phase. The weight matrix $\mathbf{W} = [w_1, w_2, b]^T$ and the input vector $\mathbf{x} = [x_1, x_2, 1]$. The cost function is defined as $C^2 = \|\hat{\mathbf{y}} - \mathbf{y}\|^2$, where $C =$

$\hat{\mathbf{y}} - \mathbf{y}$, $\hat{\mathbf{y}}$ is the expected output and \mathbf{y} is the actual output. The weight matrix is updated after each iteration by computing the gradient of the cost function $\Delta \mathbf{W} = \mathbf{C} \times \mathbf{x}^T$. The targeted values for all logic gates are defined by $(e^{i\frac{\pi}{4}}, e^{i\frac{5\pi}{4}})$ as logical “0” and $(e^{i\frac{3\pi}{4}}, e^{i\frac{7\pi}{4}})$ as logical “1”. No activation function is applied in this task, and we directly use the coherent detection results of the optical chip for classification. The numerical simulation was conducted in Python. The single complex-valued neuron is built based on an open source code⁶⁹.

Iris classification:

The inputs of Iris tasks is 4-dimensional, together with an additional bias “1”. The 4-dimensional features in Iris dataset are real-valued, corresponding to the petal length, petal width, sepal length and sepal width of three Iris species. The inputs are mapped to a total of 5 optical waves that are in phase. The output attributes of Iris task are 3-dimensional. Therefore, the weight matrix of the complex-valued layer is $\mathbf{W} \in \mathbb{C}^{5 \times 3}$. The output attributes of Iris task are 3-dimensional. Therefore, the weight matrix of the complex-valued layer is $\mathbf{W} \in \mathbb{C}^{5 \times 3}$. The target values for neurons shown in Fig 4b, 4c and 4d are $(e^{i\frac{\pi}{9}}, e^{i\frac{17\pi}{9}}, e^{i\frac{15\pi}{9}})$, $(0.6, 1.0, 1.4)$ and $(\frac{1}{2}e^{i\frac{\pi}{2}}, \frac{1}{2}e^{-i\frac{\pi}{2}}, \frac{3}{2}e^{-i\frac{\pi}{2}})$, respectively. The numerical model for complex-valued layer is built with TensorFlow with the cross-entropy cost function. The activation function applied is $M(z) = \|z\|$, where z is the complex-valued output. The training algorithm is RMSPropOptimizer.

Handwriting classification:

In the classification of dataset MNIST, we investigate different network settings for encoding and detection using various combinations of complex and real values. The encoded input signals are 4-dimensionally, with an additional bias “1”. In real-valued input encoding, the inputs are mapped to a total of 5 optical waves that are in phase. In complex-valued input encoding, the phase of the light wave in each path is deliberately modulated. The intensity and coherent detection of the chip outputs correspond to two activation functions $M(z) = \|z\|$ and $\text{ModReLU}(z) = \text{ReLU}(\|z\| + b)e^{i\theta z}$, respectively. Again, the numerical model of the multi-layer perceptron is built in TensorFlow and trained by RMSPropOptimizer with cross-entropy cost function.”

Comment 16: *Some minor questions:*

1. The results of XOR are not shown
2. When you achieve 99.3% for Iris task, how many samples are evaluated?
3. “Inner phase shifter” and “outer phase shifter” are not defined.

Answer:

1. The results of XOR is shown in Fig. 3c, with 10 iterations of training.
2. 150 samples are evaluated. We add to the manuscript as

“The numerically simulated accuracy of a single complex-valued layer is benchmarked against its real-valued counterpart. The complex-valued layer achieves an accuracy of 99.3%, which outperforms the real-valued architecture which has an accuracy of 97.3%, on the entire dataset with 150 samples.” In Line 9 on Page 13.

- We want to change the expression of “inner” and “outer” to “internal” and “external”, and define the “internal” and “external” phase shifters as “ $U(N)$ is a $N \times N$ unitary matrix represented as a product of rotation matrices $\{T_{pq}\}$ and a diagonal matrix D , such that $U(N) = \prod_{p=2}^N \prod_{q=1}^{p-1} T_{pq} D$, where T_{pq} is defined as the N -dimensional identity matrix with the elements t_{pp} , t_{pq} , t_{qp} and t_{qq} replaced by

$$\begin{bmatrix} t_{pp} & t_{pq} \\ t_{qp} & t_{qq} \end{bmatrix} = ie^{i\frac{\theta}{2}} \begin{bmatrix} e^{i\phi} \sin \frac{\theta}{2} & e^{i\phi} \cos \frac{\theta}{2} \\ \cos \frac{\theta}{2} & -\sin \frac{\theta}{2} \end{bmatrix} \quad (1)$$

where θ is defined as the internal phase shifter between two beam splitters and ϕ is the external phase shifter at the end of the MZI” in Line 19 on Page 7.

Comment 17: Some mistakes in the Supplementary

- The carrier-injection modulator is challenging to operate at 10 Gbit/s. I guess the authors want to say carrier-depletion modulation.
- Cross-section of the waveguide figure: the notation of the heater is not correct. It should be 3 um instead of 100 um. “heater” should be on top of the yellow region instead of the grey region.

Answer:

- We have revised the Supplementary correspondingly. “Therefore, with carrier-depletion modulators that have a modulation rate of 10GHz, the ONC can perform $4N^2 \times L \times 10^{10}$ MAC/s.” In Line 11 on Page 10.
- We revised the notations in Fig. S1 as

Fig. S1 | Cross-section of the waveguide. The TiN heater has length of 100 μm, width of 3 μm and thickness of 120 nm. The distance between TiN heater and the top of waveguide is 2 μm.

We hope that the extra technical details added to the manuscript, and the enclosed responses makes these points clearer, and clarify the Referee's queries about the design of the scheme and experimental details. We found the Reviewer's feedback extremely helpful – clearly indicating several places where the clarity of our results could be further enhanced. The resulting manuscript is now significantly more complete and precise, and we are very grateful for the Referee's essential contribution in instigating these improvements.

Manuscript ID:	Nature Communications manuscript NCOMMS-20-14526
Paper title:	An Optical Neural Chip for Implementing Complex-valued Neural Network
Authors:	H. Zhang, M. Gu, X. D. Jiang, J. Thompson, H. Cai, S. Paesani, R. Santagati, A. Laing, Y. Zhang, M. H. Yung, F. K. Muhammad, G. Q. Lo, X. S. Luo, B. Dong, D. L. Kwong, L. C. Kwek, and A. Q. Liu

Reply to Reviewer 2

We are grateful to the Reviewer for their candid comments. As with any large engineering project, the writing of manuscript is challenging as it involves many decisions of what details to include. Having an impartial eye is therefore extremely valuable. Here we see that our previous paper did not fully clarify the novelty and scalability of our system. We have made significant revisions to the manuscript, and document point by point responses below.

Comment 1: *The idea of computing with complex neural networks is not novel and has been investigated in the recent past in the work of Prucnal et. al. and in various reservoir computing implementations. The related increase in computational power related to real-valued approaches is also a well documented fact. The idea of training complex weights is also not new and for example was studied (albeit in simulation) in the work of Martin Fiers et. al.: Nanophotonic Reservoir Computing With Photonic Crystal Cavities to Generate Periodic Patterns and a number of other recent works (see also 10.1109/TNNLS.2018.2874571 that presents two complex-valued algorithms to training complex-valued Optical Neural Networks).*

Answer: As suggested by the Reviewer, we revise the manuscript, citing the two complex reservoir and stating their difference with neural networks as

“In reservoir computing, complex-valued reservoirs also contributes to enrich system dynamics and improve the performance^{46,47}.” In Line 17 on Page 4.

And in Supplementary information 1 in Line 1 on Page 3 as

Complex-valued reservoirs was reported in reservoir computing for enriching the system dynamics and improve the performance⁴. Corresponding training methods of the reservoirs’ read out weights were investigated⁵. However, the reservoir computing are not general neural networks, as the internal dynamics of reservoirs are uncontrollable, while we expect to be able to control the weight matrices in a neural network. Moreover, a neural network is known as a universal approximator of any mathematic function⁶, while reservoir computing has no such guarantee. ”

We gather that the Reviewer has voiced two particular concerns about how complex optical circuits differ from reservoir computation and other related past work. We would like to reassure the Reviewer that there are indeed notable differences.

In particular, reservoir computers are not general neural networks. They present a way of harnessing specific dynamics of a complex physical system¹ to scramble input information in a way that a standard neural network may require significant overhead to do. The rationale being that certain complex physical systems (e.g. those that are highly non-linear, or near phase-transition) can enable information-processing dynamics that would otherwise be very challenging to synthesize in a standard neural network.

As such, the function of a reservoir and a neural network is very different. In particular

- *Reservoirs have fixed dynamics* - A reservoir is a fixed system, representing a ‘black-box’ whose internal dynamics we cannot control and thus are not updated during the training process. In contrast, we expect to be able to have complete control over the dynamics of our optical neural network.
- *Non-universal* – A neural network is known (once made sufficiently large), to be able to approximate any mathematical functions. A reservoir, being of fixed dynamics, has no such guarantee.

The advantage of reservoir computing is that while certain non-linear systems are able to process information in exotic ways (e.g. quantum systems, liquid state machines), their internal dynamics is technologically very challenging to control. As such, reservoir computing provides a means to piggy-back off their useful information processing properties for specific case uses. Consequently, specific reservoir choices may work very well for specific problems such as frequency generator, solving Mackey-Glass equation, and the multiple superimposed oscillator (MSO) problem as demonstrated in the reference papers. However, they are not general learners.

In our context, our goal was to not to simply harness some complex number processing to perform a specialized function, but rather to build a general-purpose neural network that has built-in complex number processing. This involves new operational requirements, such as the ability to adjust the weight of interactions between each element of our complex neural network.

We hope this helps clarify the key differences between our work and that those mentioned by the review.

¹ (here complex refers to complexity of the physical system, not to be confused with it necessarily having to manipulate complex numbers)

Comment 2: *Their implementation also borrows heavily from recent work on MAC units based on MZIs. The main contributions of the work seem to be a hardware implementation tested in various application scenarios and making a more detailed comparison of complex-valued neural networks with their real-valued counterparts. Specifically, the main claim of " We tackle these issues by proposing and experimentally realizing an optical neural chip (ONC) that executes complex-valued arithmetic ..." is exactly how other coherent light-based Optical NN implementations operate.*

Answer: There are earlier works on coherent light-based optical neural network implementation, such as Shen, Y. et al. "Deep learning with coherent nanophotonic circuits." Nature Photonics, 2017, as we cited in the manuscript as

"A classical fully connected neural network has been experimentally demonstrated on an integrated silicon photonic chip^{33, 34}. Although these optical chips are based on light interference, the implemented neural network algorithms are real-valued, which forfeits the potential of complex-valued neural networks." in Line 5 on Page 4.

Although their optical chips are designed based on light interference, the implemented neural networks are based on the same real-valued framework which is commonly used by digital electronic computers.

In contrast, our optical neural chip implements a complex-valued neural network

1. Has the ability to encode the input in phase, magnitude or the both.
2. Is capable of executing on-chip coherent detection.
3. The input encoding, MAC operation and coherent detection are integrated onto a single chip.

Our ONC has proven performance advantages over real-valued counterparts in various tasks at both single-neuron and the network level. Notably, a single complex-valued neuron is able to solve certain nonlinear tasks that its real-valued counterpart is incapable of. Moreover, complex-valued networks on our ONCs demonstrate marked improvements in classification and handwriting recognition tasks. Thus, we have illustrated the potential of complex-valued optical neural networks to feature versatile representations, easy optimization and rapid learning.

Comment 3: *The authors have also not addressed the issue of how the loading phase of the data (from high speed memory?) to be trained onto their optical neural network could impact the power efficiency and speed of operation, especially as the number of MZI units and bandwidth increases.*

Answer: As suggested by the Reviewer, we add details about our loading phase of the data, in Supplementary information 5 in Line 4 on Page 10 as

“We load the phase information from the ML dataset stored in the local memory. The modulation speed of the input encoding part is limited by the modulation speed of thermo-optic modulators which is ~ 10 kHz. In our architecture, an N -dimensional input vector requires N MZIs. The scaling problem with speed and power efficiency do exist with our current proof-of-principle chip design if it scales to hundreds of modes.

To improve the power efficiency and operation speed, we consider adopting advanced modulators in future chip design, such as the high-speed (26 Gbit/s) and ultra-low-power (3.8 mW) carrier-depletion modulator ¹². We can also potentially mitigate the power scaling problem with other types of modulators, such as the ones based on phase change materials ¹³ or ultralow power MEMS phase shifters ⁴. Trade-off between the speed and power efficiency will always be present, like the case of high-speed carrier-depletion modulator which requires upgraded electrical surfaces.

From the perspective of the circuitry design, we have a substitute for the universal linear optical circuit to reduce the required number of MZIs, namely the Fast approximation design. Researchers from the machine learning community have affirmed the usefulness of Fast approximation of rotations in network constitution and training ¹⁴. The Fast design reduces the required computational resources by a large extent. The universal design consists of $N(N-1)$ training parameters, whereas the Fast design consists of only $N \log_2(N)$ matrices which is the minimum cost to allow all input coordinates to interact with each other. This design is certainly non-universal, but one can always find the approximation of the optimal weight matrix in the subspace formed by the Fast design when training a neural network, which significantly reduces the training space and eases the burden of training process.”

Comment 4: *The scaling approaches outlined by the authors towards multi-layer Optical NNs seem to introduce extensive energy efficiency challenges and operational complexity that may hit the practical limits. For example the strategy for faster operation based on free carrier modulation effects would be quite challenging to implement. For example 10GHz (or higher) signals require expensive RF I/O equipment and high speed interfaces (for example each phase shifter in each modulator would require its own 10GHz RF source signal). The other components of the system would also become more challenging to deal with e.g. multiple high speed ADCs, TIAs. Are there potential was around this?*

Answer: In our proof-of-principle design, thermo-optic heaters with a modulation speed of ~ 10 kHz are used. The whole system operates at a relatively low frequency. Nevertheless, it could potentially reach tens of GHz when the thermo-optic modulators are replaced by carrier-depletion modulators. In that case, the optical chip requires advanced electrical interface, such as the RF I/O and high-speed ADCs, TIAs, as the Reviewer suggested.

We discuss about the high-speed electronic interface in Supplementary information 2 in Line 4 on Page 5 as

“When using high-speed modulators, the optical chip requires advanced electrical interface, such as the RF I/O and high-speed ADCs, TIAs, but is achievable. A 60 GHz complete transmitter and receiver on System-in-Package (SiP) was demonstrated a few years ago²⁷. Although a 100 GHz modulator is still challenging at the moment, realizing a 10 GHz RF Transmitter and receiver along with modulators on SiP no longer presents a technical challenge. Now commercial products for complete single-chip solution for 10 GHz (VYYR2401), 60 GHz (IWR6843AoP, IWR6843) and 77 GHz (IWR1843, IWR1642) are available in the market. High-speed modulators have already successfully served in many speed-demanding applications, such as central processing unit (CPU)–memory interconnects²⁸ and data centers^{29, 30}.”

Comment 5: *The work needs very thorough proof-reading as there are numerous grammatical and other errors that some times make the manuscript challenging to follow.*

Answer: As suggested by the Reviewer, we have carefully proof-read our manuscripts to correct the grammatical errors. We hope that we have eliminated many of these issues, and thank the Reviewer for bringing them to our attention.

Comment 6: *In response to one of the earlier reviewers' questions about power efficiency, the authors claim that "once trained and configured, the system does not consume any power" (Round 1_reviewer 3 comment 4). Isn't this only the case if the weights (the modulator phases) are non-volatile. Don't you still consume power to maintain the specified phase configuration? And how much?*

Answer: We thank the Review for pointing this out. Our initial reason for writing this is under the assumption of using zero-static-power phase-change optical modulator, the required standing cost would be negligible once it's trained. The Referee is indeed correct in that electrical power is required for to maintain the specified phase configuration in our current implementation. In particular, the average electrical power required for a 2π phase shift is 70mW (14.1mA). In our new batch of design, by increasing the length of TiN heaters and digging deeper trenches to the Si layer, the required electrical power required for 2π phase shift is now reduced to 6.1mW (3.2mA, 600 Ω).

We have revised the supplementary information 2 that includes this sentence accordingly, in Lline 20 on Page 3.

“The average electrical power required for a 2π phase shift is 70mW (14.1mA). Notably in actual experiments, not all phase shifters are working at a full load. In our new batch of design, by increasing the length of TiN heaters and digging deep trenches to the Si layer, the required electrical power required for 2π phase shift is reduced to 6.1mW (3.2mA, 600 Ω). Once the chip is trained, the electrical power applied on phase shifters remain as

standing costs. It's proposed that the power consumption can be further reduced^{1, 2} by adopting zero-static-power phase change materials^{3, 4} or ultra-low power MEMS phase shifters^{5, 6}. Under this circumstance, the required standing cost would be negligible."

Comment 7: *Regarding the example on handwriting recognition, do the authors envision a feasible way to integrate the input weight matrix (784 x 4) on the same chip as their Optical Neural Network? Are there any significant computational benefits to doing this?*

Answer: We add the paragraph in Supplementary information 13 in Line 17 on Page 20 as:

"In handwriting recognition, the datasets consists of 28 × 28-pixel grayscale images. The dimensionality of the dataset is much higher than that of our optical neural chip. And it is impractical to create an optical neural network with 784 input nodes to accommodate all the 784 pixels as inputs. In order to feed our complex architecture with sufficient information while limiting the network size, we proposed to encode the high dimensional real inputs into low dimensional complex values to reduce the dimensionality. The method we demonstrate in this paper is to compress the original high-dimensional data with a complex-valued encoder (the input layer), and the input layer is implemented electrically. We also study an alternative method which is based on k-space representation²⁸. The k-space profiles are complex-valued and mostly concentrated around small k_x and k_y in the centre of the profiles, thus able to ability to reduce the number of input parameter.

The k-space is an extension of the concept of Fourier space which represents the spatial frequency information of an object in two or three dimensions. To be more specific, an image is processed by the 2D Fourier transform $c(k_x, k_y) = \sum_{m,n} e^{jk_x m + jk_y n} g(m, n)$, where $g(m, n)$ is the gray scale value of the pixel located at location (m, n) within the image. These coefficients are all complex-valued. The k-space profiles are mostly concentrated around small k_x and k_y in the center of the profiles as seen in Fig. S18. Hence it is possible to retain most of the information using only small-k components. Therefore, we can restrict the data to N coefficients with the smallest k, fulfilling the goal of decreasing the input size.

One major benefit of the k-space representation is its ability to reduce the number of input parameters hence the size of the neural network. Additionally, this method reduces the optical chip size as well as the computational resources required to for training because the dimension of the neural network does not need to accommodate all the 784-pixel values as inputs. The Fourier representations are complex-valued, which suits well with our complex-valued neural network architecture. The preprocessing procedure can be further accomplished optically by Fourier optics. We perform a numerical simulation to prove that the k-space representation works in a complex-valued neural network and compare it with a real-valued architecture. A simple activation function based on intensity detection $M(z) = ||z||$ is adopted instead of using a linear model. We test the models on

the dataset MNIST and Fashion-MNIST and report the results in Table. S3 and Table. S4. The k -space coefficients are fed into an optical neural network consisting of L layers with N neurons in each later, after which an output layer reduces the final output to 10 attributes. The real-valued architectures in comparison have input dimensions twice as large. In other words, the weight matrix for complex-valued network is $N \times N$, and that for real-valued network is $2N \times N$.”

Fig. S18 | k-space representation of images from dataset MNIST and Fashion-MNIST.

Table. S3 | Comparison of NLAf in network with L=2 (MNIST)

		Cplx-abs	Cplx-linear	Real-abs	Real-linear
K = N = 16	Train_accu	95.66%	90.71%	93.34%	87.52%
	Test_accu	95.06%	89.86%	92.34%	86.98%

Table. S4 | Comparison of NLAf in network with L=2 (Fashion-MNIST)

		Cplx-abs	Cplx-linear	Real-abs	Real-linear
K = N = 16	Train_accu	85.24%	80.60%	83.68%	77.89%
	Test_accu	84.19%	79.83%	82.38%	76.94%
K = N = 36	Train_accu	88.39%	83.69%	87.57%	80.99%
	Test_accu	86.07%	83.23%	85.70%	80.46%

Comment 8: *I find the lower most schematic in Figure 1 confusing to interpret as it does not directly map to figure above it. It might help to make it a figure of its own and caption it accordingly.*

Answer: As suggested by the Reviewer, we separate the figure and give it a label and corresponding captions as

“(c) The workflow of the ONC system. A coherent laser at 1550nm is used for generating the signal light to feed the ONC chip and a reference light to assist the coherent detection. The signal light is modulated by their magnitudes and phases according to the machine learning (ML) task. The weighted sum operation is accomplished by the light inference passively and signal light in each path is measured coherently. The measurement results are sent to the electrical interface for processing, including applying activation function and computing cost function. The ONC chip are reconfigured accordingly to the updated weight matrices. The main components (i.e., input coding, weight multiplication, and coherent detection) are integrated onto the same chip.”

We hope that the included revisions have addressed the Reviewer’s concerns over the novelty and prospects of our scheme. We found the Reviewer’s feedback extremely helpful – clearly indicating several places where we had not included enough detail about the distinction between our contribution and previous works. We are grateful that the Reviewers shares our interest in the optical (and other alternative hardware) implementation of complex neural networks. Their opinions enlightened us to think more deeply about the future of complex neural networks and the advantages of reservoir computing.

Manuscript ID:	Nature Communications manuscript NCOMMS-20-14526
Paper title:	An Optical Neural Chip for Implementing Complex-valued Neural Network
Authors:	H. Zhang, M. Gu, X. D. Jiang, J. Thompson, H. Cai, S. Paesani, R. Santagati, A. Laing, Y. Zhang, M. H. Yung, F. K. Muhammad, G. Q. Lo, X. S. Luo, B. Dong, D. L. Kwong, L. C. Kwek, and A. Q. Liu

Reply to Reviewer 3

We are grateful to the Reviewer for their affirmation to our work by “The topic of complex optical neural networks is quite intriguing and makes a lot of sense in the roadmap of optical computing. Because coherent optical signals are inherently complex numbers.” We are extremely happy to address the questions raised by the Reviewer, with regard to the the experimental demonstration of high-dimensional ML inputs and whether the complex idea work for large realistic datasets.

Comment 1: *It would be good if the author can clarify what was done optically and what was done electronically in their experiment section. To my understanding, the W_{in} and W_{out} are both done in the electronic domain, while only the middle 4×4 matrix was done optically. So how many weights are in W_{in} and W_{out} ? If W_{in} and W_{out} are a lot larger than 4×4 , then it is hard to argue the computation was done optically. Also, in theory, one can always break a bigger matrix (including W_{in} and W_{out}) into a smaller matrix and implement them optically, why didn't the author do that?*

Answer: Firstly as suggested by the Reviwer, we clarify the dimension and the implementation of the input, hidden and output layers in the manuscript as:

“The hidden layer is trained off-chip and implemented on the ONC. The input layer (784×4) and the output layer (4×10) are executed electrically.” in Line 19 on Page 15.

We then explain our reason of executing the input layer electrically. In handwriting recognition, the datasets consists of 28×28 -pixel grayscale images. The dimensionality of the dataset is much higher than that of our optical neural chip. It is impractical to create an optical neural network with 784 input nodes to accommodate all the 784 pixels as inputs. The Reviewer suggests a possible way of optically accomodating the input dataset by decomposing a large matrix into small matrices and implementing them on the optical chip step by step. We did not choose this approach for practical reasons. As each neuron only accepts 4 inputs at a time, each input sample has to be reconfigured $784/4 = 196$ times, and the 196 output will still need to be summed up electrically. Moreover, the contribution of this work is the optical implementation of the complex-valued neural network, and the key to the proof-of-principle is the optical implementation of the hidden layers which usually have a dimension several times smaller than

the input layer. Therefore, effort is put into the optical implementation of the hidden layer instead of the input layer.

We investigate an alternative method of accommodating large dimension image inputs and add the paragraph in Supplementary information 13 in Line 17 on Page 20 as

“In handwriting recognition, the datasets consists of 28×28 -pixel grayscale images. The dimensionality of the dataset is much higher than that of our optical neural chip. And it is impractical to create an optical neural network with 784 input nodes to accommodate all the 784 pixels as inputs. In order to feed our complex architecture with sufficient information while limiting the network size, we proposed to encode the high dimensional real inputs into low dimensional complex values to reduce the dimensionality. The method we demonstrate in this paper is to compress the original high-dimensional data with a complex-valued encoder (the input layer), and the input layer is implemented electrically. We also study an alternative method which is based on k -space representation⁸⁰. The k -space profiles are complex-valued and mostly concentrated around small k_x and k_y in the centre of the profiles, thus able to ability to reduce the number of input parameter.

The k -space is an extension of the concept of Fourier space which represents the spatial frequency information of an object in two or three dimensions. To be more specific, an image is processed by the 2D Fourier transform $c(k_x, k_y) = \sum_{m,n} e^{jk_x m + jk_y n} g(m, n)$, where $g(m, n)$ is the gray scale value of the pixel located at location (m, n) within the image. These coefficients are all complex-valued. The k -space profiles are mostly concentrated around small k_x and k_y in the center of the profiles as seen in Fig. S18. Hence it is possible to retain most of the information using only small- k components. Therefore, we can restrict the data to N coefficients with the smallest k , fulfilling the goal of decreasing the input size.

One major benefit of the k -space representation is its ability to reduce the number of input parameters hence the size of the neural network. Additionally, this method reduces the optical chip size as well as the computational resources required to for training because the dimension of the neural network does not need to accommodate all the 784-pixel values as inputs. The Fourier representations are complex-valued, which suits well with our complex-valued neural network architecture. The preprocessing procedure can be further accomplished optically by Fourier optics. We perform a numerical simulation to prove that the k -space representation works in a complex-valued neural network and compare it with a real-valued architecture. A simple activation function based on intensity detection $M(z) = ||z||$ is adopted instead of using a linear model. We test the models on the dataset MNIST and Fashion-MNIST and report the results in Table. S3 and Table. S4. The k -space coefficients are fed into an optical neural network consisting of L layers with N neurons in each later, after which an output layer reduces the final output to 10 attributes. The real-valued architectures in comparison have input dimensions twice as

large. In other words, the weight matrix for complex-valued network is $N \times N$, and that for real-valued network is $2N \times N$.”

Fig. S18 | k-space representation of images from dataset MNIST and Fashion-MNIST.

Table. S3 | Comparison of NLAf in network with L=2 (MNIST)

		Cplx-abs	Cplx-linear	Real-abs	Real-linear
K = N = 16	Train_accu	95.66%	90.71%	93.34%	87.52%
	Test_accu	95.06%	89.86%	92.34%	86.98%

Table. S4 | Comparison of NLAf in network with L=2 (Fashion-MNIST)

		Cplx-abs	Cplx-linear	Real-abs	Real-linear
K = N = 16	Train_accu	85.24%	80.60%	83.68%	77.89%
	Test_accu	84.19%	79.83%	82.38%	76.94%
K = N = 36	Train_accu	88.39%	83.69%	87.57%	80.99%
	Test_accu	86.07%	83.23%	85.70%	80.46%

Comment 2: *It is interesting to see that the complex neural networks idea starts working for the benchmarks discussed in the paper. However, all these are still considered toy tasks, and it is not very convincing to me that complex NN really works better than real-valued NN. There are better “real value” based neural networks that perform much better than the results referenced in the paper (for example, CNN can achieve >99.8% accuracy on MNIST). Therefore, it would be more interesting to see if the authors can take a state of the arts real-valued model, and improve it with complex numbers. Moreover, it would be good to see if complex NN can really work on some more realistic datasets, for example, Imagenet or CIFAR tasks. Experimental demonstration is not needed for those bigger datasets, just numerical result should be sufficient.*

Answer: Reviewer pointed out, some advanced real-valued CNN (Ciregan, Dan, Ueli Meier, and Jürgen Schmidhuber. "Multi-column deep neural networks for image classification." CVPR, 2012) can achieve a 99.8% accuracy on MNIST, but it has a much higher parameter budget of ~38580. As suggested by the Reviewer, we modified the advanced convolutional neural network (CNN) and applied it to CIFAR10 and SVHN task to check if complex-valued CNN could improve over its real-valued counterparts. The details are included in the Supplementary information 14: Complex CNN and RNN for realistic datasets in Line 10 on Page 22.

“We built a complex model that consists of a convolution layer (depth of 20, kernel of 6×6 and stride of 1), a max-pooling layer (kernel of 2×2), followed by another convolution layer (depth of 50, kernel of 6×6 and stride of 1) and two fully-connected layers. The loss function is negative log likelihood(NLL) loss

$$C = -\frac{1}{n} \sum_x y^T \ln(a^L) == -\frac{1}{n} \sum_x \sum_{k=1}^K y_k \ln(a_k^L)$$

where y is the desired output and a^L is the output of the model. The logarithmic probability in a neural network is easily achieved by adding Log-SoftMax in the last layer of the network. The training algorithm we use is the simplest Stochastic Gradient Descent (SGD) with a learning rate of 0.01, a momentum of 0.9, and a batch size of 64. Both complex-valued CNN and real-valued CNN model are tested. The testing accuracies of the complex-valued CNN and the real-valued CNN on the dataset CIFAR10 after 50 training epochs are 71% and 69%, respectively. The same network architecture on the dataset SVHN obtains a 91% accuracy with complex values and 89% with real values. More comprehensive study on advantages of complex CNN were reported in C. Trabelsi, et.al, “Deep complex networks”.

In addition, we test a complex-valued RNN and its modification complex-valued Gated Recurrent Unit (complex GRU) against real-valued RNN on the human motion prediction task²⁸. We use the Human 3.6M (H3.6M) dataset²⁹, which includes actors performing 15 varied activities such as walking, eating and smoking. We evaluate the Euclidean distance between our prediction and the ground-truth in angle space. Increasing time horizons from 80ms to 400 ms are investigated. The comparison of performance (via mean angle error) is shown in Table. S3 with the lowest error underlined. The lower the error, the better the performance of the network. The column of complex GRU is reported by M .

Wolter, et.al³⁰. The dimensions of real RNN, complex RNN and complex GRU are 1024, 512 and 512, respectively. It is observed that in most of the cases, the complex-valued architectures have better performances than the real-valued ones. And the complex GRU perform the best.”

Table. S3 | Complex RNN, Complex GRU vs. Real RNN on H3.6m dataset (via mean angle error)

Action	Real RNN (1024)				Complex RNN (512)				Complex GRU (512)			
	80ms	160ms	320ms	400ms	80ms	160ms	320ms	400ms	80ms	160ms	320ms	400ms
Walking	0.497	0.620	0.747	0.802	0.370	0.529	0.742	0.809	0.334	0.473	0.690	0.773
Eating	0.389	0.489	0.679	0.835	0.301	0.426	0.649	0.792	0.277	0.411	0.643	0.791
Smoking	0.584	0.776	1.143	1.199	0.433	0.667	1.030	1.171	0.391	0.630	1.039	1.123
Discussion	0.604	0.858	1.077	1.153	0.450	0.733	1.003	1.066	0.456	0.773	1.031	1.092
Directions	0.709	0.898	0.881	0.979	0.520	0.688	0.850	0.958	0.502	0.680	0.843	0.938
Greeting	0.829	1.055	1.370	1.530	0.632	0.902	1.277	1.446	0.610	0.891	1.257	1.410
Phoning	0.755	1.073	1.567	1.695	0.621	1.122	1.523	1.656	0.600	1.128	1.562	1.722
Posing	0.740	0.847	1.346	1.590	0.679	0.945	1.438	1.659	0.612	0.880	1.382	1.619
Purchases	0.773	0.948	1.268	1.351	0.742	0.881	1.168	1.230	0.728	0.894	1.141	1.222
Sitting	0.832	1.037	1.387	1.575	0.579	0.797	1.218	1.429	0.547	0.768	1.185	1.372
Sitting down	1.009	1.301	1.676	1.850	0.609	0.941	1.372	1.564	0.603	0.961	1.409	1.585
Taking photo	0.561	0.775	1.026	1.145	0.342	0.589	0.920	1.069	0.334	0.574	0.893	1.026
Waiting	0.573	0.803	1.214	1.386	0.405	0.656	1.090	1.265	0.389	0.628	1.048	1.223
Walking dog	0.720	0.958	1.223	1.334	0.629	0.913	1.239	1.396	0.652	0.954	1.401	1.484
Walking together	0.475	0.670	0.871	0.919	0.351	0.573	0.826	0.895	0.336	0.562	0.791	0.873
Average	0.670	0.874	1.165	1.290	0.511	0.757	1.090	1.227	0.491	0.747	1.087	1.217

We are grateful to the reviewer for their great interest in our work and for their time in assessing our manuscript. They raised very profound questions and inspired us to think deeply about the details of our scheme.

Reviewers' Comments:

Reviewer #1:

Remarks to the Author:

The authors implement a photonic vector-matrix multiplexer using the MZI meshes-based architecture proposed by the MIT group (Shen, Y. et al. Deep learning with coherent nanophotonic circuits. Nature Photonics 11, 441 (2017)). The main contribution of this work is to extend the photonic vector-matrix multiplication to the complex domain.

Complex-valued neural networks have some advantages over real-valued neural networks. Indeed, implementing complex-valued neural networks with conventional computers can be computationally expensive. However, it needs to be pointed out that, it is also not free to implementing complex-valued neural networks with photonics. You have to pay hardware costs and additional energy consumption. First, you need to double the number of ADCs and memories to implement a complex weight. The overhead of ADCs and memories in terms of price and energy is a serious concern in photonic neural networks in general. The speed of loading data from memory is the speed bottleneck of photonic neural networks. Second, you also need additional heaters to implement a complex weight. This corresponds to additional loss and noise, which may weaken the theoretical advantages in terms of accuracy.

Key information provided in the manuscript, including the chip (for example, whether there are coherent detectors on-chip or not), and how to implement the Classification of Dataset Iris test and Handwriting Recognition test, is very vague. Therefore, it is hard to conclude that the complexed-valued neural network chip would perform better than real-valued neural networks using photonic hardware. These are some of the reasons:

1. In Classification of Dataset Iris, the authors claim the advantage of complex-valued photonic neural networks is accuracy. However, the authors didn't compare the accuracy of the complex-valued photonic neural network over a real-valued one. The authors only compared the accuracy numerically. The accuracy improvement is 2% over only 150 samples. As pointed out earlier, you need additional components to implement a complex weight, which would accumulate additional physical noises. So with photonic hardware, a complex-valued neural network may not always achieve higher accuracy.
2. In Handwriting Recognition, the authors mentioned the complex-valued algorithm has ~10% higher accuracy than real-valued algorithms, which is significant. However, the reason for higher accuracy is not clear. I notice that the real-valued neural network only has an accuracy of 82%, which is significantly lower than those demonstrated using conventional computers. What's the reason for such low accuracy? Is it due to the algorithm or hardware? Why didn't the author show the numerical results as they did in the Classification of Dataset Iris?
3. It is very unclear what are the neural network models (layers, neuron numbers of each layer, the activation function and detection approaches used) under so many comparison conditions. It is very unclear which part of the neural network is implemented with the photonic chip, and which part is implemented with electronics. The missing information makes the comparisons not that convincing.

Besides, although the structure of the paper is fine, the details are not presented logically, and thus are hard to follow. Several grammatical errors also make the manuscript challenging to follow.

Reviewer #2:

None

Reviewer #3:

Remarks to the Author:

The authors have addressed my major concerns, and I recommend publishing this paper.

Manuscript ID: Nature Communications manuscript NCOMMS-20-14526A
Paper title: An Optical Neural Chip for Implementing Complex-Valued Neural Network
Authors: H. Zhang, M. Gu, X. D. Jiang, J. Thompson, H. Cai, S. Paesani, R. Santagati, A. Laing, Y. Zhang, M. H. Yung, Y. Z. Shi, F. K. Muhammad, G. Q. Lo, X. S. Luo, B. Dong, D. L. Kwong, L. C. Kwek, and A. Q. Liu

Reply to Reviewer 1

We are grateful to the Reviewers for their constructive comments, and their recognition of our contribution of on-chip implementation of complex-valued optical neural networks (ONN). We also appreciate the contributions from all reviewers in stimulating a better piece of work. The Reviewer 1 still had several concerns about the detailed design and experimental realization of the scheme. We are happy to address all such concerns below. We thank the Reviewer 1 for their input, it has catalyzed multiple improvements to our manuscript.

Comment 1: *It needs to be pointed out that, it is also not free to implementing complex-valued neural networks with photonics. You have to pay hardware costs and additional energy consumption. First, you need to double the number of ADCs and memories to implement a complex weight. Second, you also need additional heaters to implement a complex weight. This corresponds to additional loss and noise, which may weaken the theoretical advantages in terms of accuracy.*

Answer 1.1 For the number of ADCs and memories. The number of phase shifters required to implement a complex weight is **exactly the same** as the number required for a real one, as referred to the decomposition methods in the paper by Reck, Michael, et al. "Experimental realization of any discrete unitary operator". Therefore, the memories occupied by the phase values and the loading time also remain the same. The number of physical ADCs is not doubled in our chip design, but we do have to double the number of measurements for coherent detection. The comparison of the complex and real networks was shown on Page 19 as

*"We compare the costs to optically implement the complex and real neural network from three aspects: the input encoding, the weight multiplication and the detection methods. **Real encoding also requires the control of the relative phases between MZIs**, otherwise the output would be fluctuating with accumulated phase variability. In contrast, **coherent encoding uses the same number of components but contains both magnitude and phase information.** Using the coherent encoding, the input data for the hidden layer has double dimensions. Besides, both the real and complex networks require the modulation on all the internal and external phase shifters for the implementation of a real-valued matrix on a linear optical circuit. The only additional cost of complex networks is that coherent detection is more expensive than intensity detection, requiring doubled amount of measurements. However, even if we only perform the real-valued detection (**Fig. 6e**), the*

complex-valued-encoding algorithms maintain an accuracy of 88.5%, which is still significantly higher than the real-valued algorithm (82.0%).”

Answer 1.2 For the additional heater to implement a complex weight. In fact, the number of phase shifters required in implementing a complex weight matrix is **exactly the same** as a real weight matrix referring to the paper by Reck, Michael, et al. "Experimental realization of any discrete unitary operator". The specific reasons are as follows: A real-valued weight matrix is a special case of a complex-valued weight matrix with the relative phase of elements constrained to **either 0 or π** . For the simplest case of a single MZI (rotation matrix in Equation 1), the real-valued property requires $\phi = 0, \pi$. Notably, the phase shifter on silicon chip is not fabricated such that this phase is automatically always 0. Instead it generally must be **calibrated using electrical power**. Besides, real networks also require coherent detection to determine whether the output is positive or negative. There is also **no difference in the number of heaters required for implementing a complex/real matrix in linear optical circuit**, as referred to the decomposition method in the reference paper. The example of decomposing a 4-mode circuit in Appendix A shows that all internal and external phase shifters should be reconfigured for both complex and real weights. We clarify the point on Page 19 as

“Besides, both the real and complex networks require the modulation on all the internal and external phase shifters for the implementation of a real-valued matrix on a linear optical circuit.”

Comment 2: *Key information provided in the manuscript, including the chip (for example, whether there are coherent detectors on-chip or not), and how to implement the Classification of Dataset Iris test and Handwriting Recognition test, is very vague.*

Answer: We thank the Reviewer for pointing out the unclear part in our descriptions of the chip and task implementation. We add details correspondingly.

Answer 2.1 For the detailed description of the chip. We describe the **working principles of our chip from three parts, the input encoding, the weights implementation and the output detection in the section “Design and Fabrication”** from Page 6 to Page 8. As pointed out by the reviewer, we did not use a specific coherent detector like the off-chip 90-degree optical hybrid. Instead, our coherent detection is implemented on-chip by individually measuring the intensities linked with the $\cos\phi_s$ and $\sin\phi_s$ respectively. The principle of our on-chip coherent detection is revised on Page 8 as

*“Our integrated chip is capable of performing both intensity detection and coherent detection (refer to **Methods** and **SI. 7** for details). **The goal of coherent detection is to determine the phase angle ϕ_s between the reference light and signal light.** By connecting photodiodes at both outputs in a balanced way, the obtained output current is $I_I \propto 2A_s A_l \cos\phi_s$, where A_s and A_l are the magnitudes of the signal and the reference light. Similarly, by adding a phase shift of $\pi/2$ to the reference light, the output current is $I_Q \propto$*

$2A_s A_l \sin \phi_s$. The ϕ_s is then determined from the ratio of I_I and I_Q , which also eliminates the physical noises from the optical components. The choice of detection method is determined by the activation function. Intensity detection is naturally applied for the activation function $M(z) = \|z\|$, whereas coherent detection is applied for the activation function $\text{ModReLU}(z) = \text{ReLU}(\|z\| + b)e^{i\theta_z}$."

Answer 2.2 For the Iris classification. We have adjusted the narrative logic and provided a more complete description about the testing on Page 12 as

"The whole dataset with 150 samples is divided into training set and testing set by a ratio of 0.75:0.25. The weights are trained only on the training set, based on the same numerical model as in the logic gate task. As the bias is included in the input vector as a constant entry, an additional variable is assigned to the weight matrix to make the bias trainable. Thereby for the single layer with 3 neurons, the input vector $x \in \mathbb{R}^{1 \times 5}$ and the weights $W \in \mathbb{C}^{5 \times 3}$. In chip implementations, the real input vector x is encoded into the magnitude of optical waves, while keeping their phases identical. The neuron output y is complex-valued and detected coherently. The input vectors and trained weights are numerically decomposed to exact values of the respective phase shifters (see examples in SI. 5, 6). Then the specific electric power for each phase shifter is computed by its calibration curve (SI. 4). We configure our ONC with trained complex weights and show the neuron outputs of the optical chip in Figs. 4b, 4c and 4d."

Answer 2.3 For the handwriting classification. The details of the building, training and implementation of the multi-layer perceptron model are added on Page 17 as

"The dataset is split into training set and testing set. Our model is trained on the entire training set, and 200 samples in the testing set are used to validate the trained model on chip. As shown in Fig. 6a, the network consists of an input layer W^{in} , a hidden layer W and an output layer W^{out} . The corresponding numbers of neurons in the three layers are 4, 4 and 10, respectively. The 28×28 grayscale image is stretched into a 784×1 vector and compressed by the input layer into 4 outputs, which are fed to the 4×4 hidden layer. The output layer maps the 4 outputs of the hidden layer to 10 classes, representing digits from 0 to 9. The numerical model is built in TensorFlow and trained by RMSPropOptimizer, at a learning rate of 0.005, training epochs of 100 and batch size of 100. The activation function used for the real model is ReLU and for the complex model we use ModReLU (see SI. 14). The hidden layer is implemented on the ONC, while the input layer (784×4) and the output layer (4×10) are executed electrically. Theoretically, the input and output layers are implementable with our ONC by decomposing the large matrix into small ones. Considering the practical workload and the same principles, we only focused on the proof-of-principle implementation of the hidden layer."

Comment 3: In Classification of Dataset Iris, the authors claim the advantage of complex-valued photonic neural networks is accuracy. However, the authors didn't compare the accuracy of the

complex-valued photonic neural network over a real-valued one. The authors only compared the accuracy numerically. The accuracy improvement is 2% over only 150 samples. As pointed out earlier, you need additional components to implement a complex weight, which would accumulate additional physical noises. So with photonic hardware, a complex-valued neural network may not always achieve higher accuracy.

Answer 3.1 For the advantage of the complex neural network and additional physical noises.

We address these concerns from the following aspects: (a) The complex/real photonic neural network with same structure require exactly the same number of physical components, as also indicated in **Answer 1.2**. (b) We emphasize that the real architecture that we compare is 3-layered, as previously referenced to [58][59]. We did not compare to a single real layer because such a network can only form linear partitions on the data set. To achieve the same classification function in the Iris task, the conventional real model requires a 3-layer architecture, which is more costly than our single layer complex network. Therefore, our physical noises are lower than the real network, and the accuracy is not compromised. (c) We analyze the reason of the advantage of the complex model over its real counterpart. The revised paragraph about the Iris task is shown on Page 14 as

*“We benchmark the single complex layer against 3-layer real network that is commonly required for Iris classification in photonic implementations of neural networks^{58, 59}. Our complex layer has a simulated accuracy of 99.3% (see **Fig. S13**) and a chip testing accuracy of 97.4%. Whereas for the 3-layer real network, despite its comparable simulated accuracy of 97.3%, would experience a large decrease in accuracy due to the multi-layer cumulative error during physical implementation (see **Fig. S16**). Although the improvement in simulated accuracy is not evident, our complex model obtains high accuracy in physical implementation using fewer layers and neurons, thereby avoiding excessive cumulative errors.”*

Answer 3.2 For the additional new-added nonlinear experiment. Our complex layer has great advantages in learning nonlinear patterns. Whereas, the advantage is less pronounced in a dataset which contains linear-separable classes. We conduct another experiment on completely nonlinear datasets (the Circle and the Spiral) to reinforce our claims about advantages of complex network, with hugely improved accuracy and supporting analytical arguments. The formation of nonlinear decision boundaries in complex model is visualized, in comparison to that of its real counterpart. The advantages can also be viewed from the significant improvement in numerical and experimental accuracies. We also discuss the underlying principle of how complex models form the nonlinear decision boundaries, in the section “**Classification of nonlinear datasets Circle and Spiral**” on Page 14. The results are shown in **Fig. 5**. Here, we attach the relevant content and Fig. 5 as

Classification of nonlinear datasets Circle and Spiral – Here, we highlight the capability of complex networks in forming nonlinear decision boundaries, in comparison to their real counterparts. Two nonlinear datasets are studied, namely the Circle and the Spiral. The dataset

visualization, real/complex model for comparison and chip detection results are shown from left to right in **Fig. 5a**. Both datasets are entangled and linearly inseparable. They have two real-valued inputs and two classification classes. For each task, a complex model and an equivalent (same number of layers and neurons) real model are compared. For the Circle, a single complex/real layer with two neurons is adopted. Intensity detection is performed at the output ports of the chip. For the Spiral, a two-layer network is adopted. The 1st layer has four neurons and are set to complex/real value for comparison. Intensity detection is performed at this layer, which is equivalent to an activation function $M(z) = \|z\|$. The 2nd layer is a real-valued linear mapping from the 4 hidden nodes to the 2 output nodes. The classification results can be interpreted from the chip outputs $y_{1,2}$ by a simple manner of *Argmax*: if $y_1 \geq y_2$, the samples belong to the “blue” class, otherwise if $y_1 < y_2$, they belong to the “pink” class.

In the binary classification, the decision boundary partitions the underlying vector space into two regions, one for each class. The decision boundaries of the real/complex model and chip validated results of the Circle and Spiral are shown in **Figs. 5b** and **5c**, respectively. As observed, the decision boundaries of real models are formed by straight lines, while those of complex models are highly nonlinear curves that exactly match the entangled shape of the datasets. The complex model is also appreciably superior in the classification accuracy. The complex model achieves simulated accuracy of 100% in both datasets, far exceeding the 52% and 89% by the real model for the Circle and Spiral, respectively. In the chip implementation, we scan both inputs x_1 and x_2 from -1 to 1 with a step size of 0.1. A total 441 input sets are tested. The black wires as marked are our expected decision boundaries while the white wires are experimental ones, from which we can easily tell which points are incorrectly classified. The chip testing accuracies are 98% for the Circle and 95% for the Spiral. The boundaries of the complex model are theoretically smooth, while the visualized resolution is reduced by the input interval in experiment.

The theoretical decision boundaries can be derived. Suppose we have a single layer with two neurons, the outputs of the real model are

$$\begin{bmatrix} y_1 \\ y_2 \end{bmatrix} = \begin{bmatrix} w_{11} & w_{12} \\ w_{21} & w_{22} \end{bmatrix} \begin{bmatrix} x_1 \\ x_2 \end{bmatrix} + \begin{bmatrix} b_1 \\ b_2 \end{bmatrix} \quad (2)$$

where the $w_{11,12,21,22}$ and $b_{1,2}$ are real-valued, $x_{1,2}$ are real inputs and $y_{1,2}$ are the outputs. The decision boundary is derived by solving the equation $\|y_1\| = \|y_2\|$. Thereby, the decision boundary is formed by two straight lines:

$$\begin{cases} l_1: (w_{11} - w_{21})x_1 + (w_{12} - w_{22})x_2 + (b_1 - b_2) = 0 \\ l_2: (w_{11} + w_{21})x_1 + (w_{12} + w_{22})x_2 + (b_1 + b_2) = 0 \end{cases} \quad (3)$$

In the complex model, we replace the weight matrices by $w_{jk} = p_{jk} + iq_{jk}$, and $b_j = m_j + in_j$, where $j, k = 1, 2$. By solving $\|y_1\| = \|y_2\|$, a nonlinear decision boundary is formed by

$$\begin{aligned} & (p_{11}x_1 + p_{12}x_2 + m_1)^2 + (q_{11}x_1 + q_{12}x_2 + n_1)^2 = \\ & (p_{21}x_1 + p_{22}x_2 + m_2)^2 + (q_{21}x_1 + q_{22}x_2 + n_2)^2 \end{aligned} \quad (4)$$

which can be simplified to a binary quadratic equation:

$$Ax_1^2 + Bx_2^2 + Cx_1x_2 + Dx_1 + Ex_2 + F = 0 \quad (5)$$

The Equation (5) can form various curves, such as parabola, circle, ellipse and hyperbola, with different parameters $A-F$, which can be learned from training data. For nonlinear data distributions, the complex number network shows a strong learning capability and achieves a high classification accuracy.

Fig. 5 | Nonlinear decision boundaries in complex-valued network. (a) Two nonlinear datasets, the Circle and the Spiral, are investigated. Both datasets have two-dimensional real-valued inputs and two classes. For the Circle, a single complex/real layer with two neurons is adopted. Intensity detection (shown by the PDs) is performed in the chip outputs. For the Spiral, a two-layer network is adopted. The 1st layer has 4 neurons that are set to complex/real for the comparison. Intensity detection is performed to the 1st layer. The 2nd layer is a real-valued linear mapping between the hidden and output nodes. The output distribution by the chip in experiment is shown, from which the classification results can be interpreted. (b) The subfigures from left to right are the decision boundaries of the real/complex model and chip validated results in classifying the Circle. In chip implementation, we scan both inputs x_1 and x_2 from -1 to 1 by a step of 0.1. A total 441 input sets are tested. The black boundaries are expected results. Smoother boundary will be obtained if the input space is scanned denser. The same results of the Spiral are shown in (c).

Comment 4: *In Handwriting Recognition, the authors mentioned the complex-valued algorithm has ~10% higher accuracy than real-valued algorithms, which is significant. However, the reason for higher accuracy is not clear. I notice that the real-valued neural network only has an accuracy of 82%, which is significantly lower than those demonstrated using conventional computers. What's the reason for such low accuracy? Is it due to the algorithm or hardware? Why didn't the author show the numerical results as they did in the Classification of Dataset Iris?*

Answer 4.1 For the reason of higher accuracy. The reason for the higher accuracy of the complex neural network comparing to the real one is: (a) the complex weight matrix has doubled capacity (double the number of effective real-valued free parameters), although they use the same chip with same number of phase shifters. (b) the complex hidden layer receives doubled dimension encoding the input data (by real and imaginary part). That is also why we conduct ablation studies by the five scenarios. We intend to see whether the improvement is only due to the doubled input dimension or whether the complex matrix itself also contributes. Our conclusion is that both factors contribute. (c) The complex matrix itself have stronger learning capability as we discovered from the five scenarios. We added a sentence to explain the reason and our motivation of those comparisons on Page 17 as

“The complex model is superior to the same-dimensional real model, with a significant advantage of about 10%, partly because it receives more information in both magnitude and phase from the previous layer. Ablation studies are conducted to verify the strength of complex-valued weight matrix itself and the effect of the encoding and detection methods in optical realization, by implementing the complex model on our ONC in the following ways:”

Answer 4.2 For the reason of low accuracy compared to conventional computers. The reason for the low accuracy of 82% is the limited amount of training variables. Our accuracy of 82% is obtained by a real-valued multi-layer perceptron with merely a 4×4 hidden layer. Whereas the high accuracies on electronic computer are achieved by much more neurons (e.g. 98.2% by MLP 784-512-10) or advanced convolutional neural network (CNN) (e.g., 99.8% at a parameter budget

of ~ 38580 , see the paper by Ciregan, et.al. "Multi-column deep neural networks for image classification." CVPR, 2012). We can see that the gap in accuracy is caused by the huge difference in parameter budget. The objective of the experiments is not to achieve the state-of-the-art performance, which need a very large network. Our purpose is to compare our developed photonic complex-values optical network (currently small in size) with its real-valued counterpart.

Answer 4.3 For the requirement of showing numerical results. We agree with the reviewer that further elaborating using new data would be helpful for illustrating the handwriting tasks. Therefore, we add the confusion matrix of the on-chip testing results of the complex and real model in Figs. 6c and 6d. From the confusion matrix, one can get detailed information about the prediction of each sample. Here, we also show the experimental records of 10 samples about their inputs and outputs of the complex hidden layer in Appendix B. For each sample, the original 784 (28×28) entries are compressed to 4 complex inputs as shown in Table 1. The 4 complex outputs after being weighted by the optical chip, are listed in Table 2. They are then transformed by a 4×10 output layer, and returned as a 10×1 vector in Table 3. The index of the maximum value in the entry of each sample corresponds to its category, and are marked with green color.

The numerical results of MNIST classification are not visualized as the same type of graph in the *Iris* classification, because their network structures are all different. The *Iris* classification employed a single complex layer. Thus, it is intuitive to look at how the direct neuron outputs are distributed on the complex plane. Whereas, MNIST classification uses a 3-layer MLP (an input layer, a single hidden layer and an output layer) and the output is mapped to category by the function Argmax. Argmax finds the index that gives the maximum value from the output vector, which is in fact the class with the largest predicted probability. Concretely, the numerical output of MNIST is interpreted by finding the index of the element with maximum value in a 10×1 vector.

Fig. 6 | (c) The confusion matrix of testing samples under the complex model. Each column of the matrix represents the instances in a predicted label while each row represents the instances in a true label. The diagonal elements represent the number of samples that are correctly predicted. (d) The confusion matrix under the real model.

Comment 5: *It is very unclear what are the neural network models (layers, neuron numbers of each layer, the activation function and detection approaches used) under so many comparison conditions. It is very unclear which part of the neural network is implemented with the photonic chip, and which part is implemented with electronics. The missing information makes the comparisons not that convincing.*

Answer 5.1 For the details of network models. The basic neural network model is a multi-layer perceptron with an input layer, a single hidden layer and an output layer. **The architecture is fixed for all the five comparison scenarios.** The description of neural network model is added in the revised manuscript on Page 17 as

For layers and neuron numbers: *“As shown in Fig. 6a, the network consists of an input layer W^in , a hidden layer W and an output layer W^{out} . The corresponding neuron numbers in the three layers are 4, 4, 10, respectively. The 28×28 grayscale image is stretched into a 784×1 vector and compressed by the input layer into 4 inputs to feed the 4×4 hidden layer. The output layer maps the 4 outputs to 10 classes representing digits from 0 to 9.”*

and for implementation:

“The hidden layer is implemented on the ONC, while the input layer (784×4) and the output layer (4×10) are executed electrically.”

Answer 5.2 For the activation functions and detection approaches. The activation function and detection method applied to each of the five scenarios are: (a) *ModReLU*, coherent detection; (b) *ModReLU*, coherent detection; (c) $M(z)$, intensity detection; (d) $M(z)$, intensity detection; (e) *ReLU*, intensity detection. We describe briefly about the relationship of activation functions and detection approaches on Page 18 as

“The intensity and coherent detection of the chip outputs correspond to the activation functions $M(z) = \|z\|$, and $ModReLU(z) = ReLU(\|z\| + b)e^{i\theta z}$, respectively.”

In summary, we have added a detailed description of the design and optical implementation of our complex-valued neural network and classified that number of optical components required by the complex neural network is equal to the same layered real ones. To highlight its advantages, we explained the reason for the inconspicuous improvement of accuracy to be the dataset itself, and added a new case of nonlinear classification, showing a marvelously improved accuracy. We hope these answers and efforts could provide the reviewer a more in-depth image of the advantages of our complex-valued neural network in increased accuracy and lower hardware costs.

The resulting manuscript is now significantly completer and more precise, and we are very grateful for the Referee’s essential contribution in stimulating these improvements.

Appendix A

The numerical example of decomposition of real/complex weights

For a randomly generated unitary matrix

$$u = \begin{bmatrix} -0.2160 + 0.6751i & -0.0457 - 0.5967i & -0.1928 - 0.1184i & 0.2418 - 0.1725i \\ 0.3949 + 0.0266i & 0.6247 + 0.1079i & -0.5114 + 0.2750i & 0.2937 - 0.1345i \\ -0.2726 + 0.1448i & -0.2726 + 0.1448i & -0.3300 + 0.0143i & -0.8076 - 0.3113i \\ -0.4558 + 0.1945i & -0.2262 + 0.3772i & -0.0065 + 0.7089i & 0.2322 - 0.0665i \end{bmatrix}$$

the decomposed phase shifter values are

θ_1	θ_2	θ_3	θ_4	θ_5	θ_6				
1.5756	2.0259	1.3025	1.1914	2.0265	1.1140				
ϕ_1	ϕ_2	ϕ_3	ϕ_4	ϕ_5	ϕ_6	α_1	α_2	α_3	α_4
-2.8467	-2.4075	1.1554	-0.5432	1.4397	-0.0850	2.3685	-1.3354	0.6266	-3.6852

Similarly, for a randomly-generated real-valued matrix (orthogonal as analogue to unitary)

$$w = \begin{bmatrix} -0.5024 & 0.2622 & -0.8127 & 0.1357 \\ -0.5278 & 0.2240 & 0.2694 & -0.7737 \\ -0.4496 & -0.8919 & -0.0017 & 0.0479 \\ -0.5165 & 0.2926 & 0.5167 & 0.6169 \end{bmatrix}$$

the decomposed phase shifter values are

θ_1	θ_2	θ_3	θ_4	θ_5	θ_6				
1.0527	0.6161	2.8111	1.3132	0.8666	1.4323				
ϕ_1	ϕ_2	ϕ_3	ϕ_4	ϕ_5	ϕ_6	α_1	α_2	α_3	α_4
-1.7688	-5.4621	-1.8788	-3.9962	-5.3690	-6.2832	2.8133	3.2549	-0.9842	2.1574

As observed, all the internal and external phase shifters should be reconfigured for both complex and real weight. Unless when intensity detection is applied to the network, the phases ϕ_1, ϕ_4, ϕ_6 can be treated as global phases.

Appendix B

The experimental records of 10 samples of the complex hidden layer

Table 1: Inputs of the complex hidden layer

Index	Input #1		Input #2		Input #3		Input #4	
	ρ	θ	ρ	θ	ρ	θ	ρ	θ
1	8.49	2.90	57.86	-1.37	23.09	2.23	7.42	2.89
2	19.08	-0.57	90.11	0.65	33.92	2.37	11.84	1.59
3	6.42	1.98	10.16	0.34	12.36	-1.99	0.00	0.00
4	4.00	-1.67	14.27	0.28	32.93	0.76	0.00	0.00
5	12.49	-2.11	10.07	-1.28	18.09	-0.16	5.68	-1.93
6	16.86	2.25	13.12	-0.02	12.05	-2.08	0.00	0.00
7	3.18	-2.28	11.19	-2.53	18.37	0.17	15.24	3.00
8	10.05	-0.01	16.15	-0.92	23.48	-1.18	11.53	2.58
9	27.23	-0.59	20.64	1.01	27.19	-0.06	14.69	-2.18
10	7.42	2.63	42.01	-1.51	28.24	-0.16	15.84	-3.09

Table 2: Outputs of the complex hidden layer

Index	Output #1		Output #2		Output #3		Output #4	
	ρ	θ	ρ	θ	ρ	θ	ρ	θ
1	5.19	-1.97	15.79	0.23	21.75	-0.23	8.82	-2.95
2	19.41	-2.94	26.86	1.77	17.41	1.83	16.22	-0.84
3	3.92	-1.11	5.58	2.18	6.53	1.24	1.70	1.77
4	11.07	1.72	2.16	0.37	3.46	-2.64	5.80	-2.08
5	10.88	0.63	5.65	-1.30	2.85	2.88	5.90	-2.40
6	4.76	-1.20	8.12	2.20	10.89	1.00	4.53	2.45
7	9.78	-0.35	3.91	-1.70	7.68	-1.83	5.62	-3.11
8	11.89	-0.75	5.61	0.91	1.98	-0.80	3.42	2.02
9	17.60	0.70	1.60	-1.40	10.43	-2.73	8.29	-0.92
10	15.99	-0.07	7.59	-0.26	11.09	-0.63	12.45	3.02

Table 3: Mapping to categories

index	#0	#1	#2	#3	#4	#5	#6	#7	#8	#9
1	-13.72	-29.35	-15.36	-8.08	-17.85	-10.01	-39.15	2.01	-14.46	-8.86
2	14.75	12.47	27.00	19.53	-17.22	8.13	18.44	-19.22	10.85	-12.39
3	-13.81	-0.31	-7.73	-7.50	-8.86	-7.23	-7.81	-7.63	-7.05	-8.81
4	9.79	-10.37	0.88	-0.27	-3.80	1.57	0.00	0.61	2.56	-0.41
5	-11.33	-20.33	-11.29	-14.57	-3.55	-10.64	-10.23	-7.32	-10.37	-7.24
6	-24.23	-8.04	-16.49	-15.42	-19.90	-18.71	-20.48	-13.95	-14.41	-17.46
7	-19.95	-18.92	-19.84	-15.82	-6.47	-10.62	-16.56	-11.76	-11.67	-10.08
8	-24.05	-9.94	-13.76	-11.23	-6.07	-10.85	-14.99	-13.33	-10.86	-2.91
9	4.46	-8.79	4.75	-5.83	10.42	8.39	14.36	-6.83	6.24	3.55
10	-34.25	-38.73	-33.62	-28.63	-17.25	-25.19	-40.73	-15.50	-21.14	-11.54

Reviewers' Comments:

Reviewer #1:

Remarks to the Author:

The authors have spent significant efforts in addressing all my comments and questions. I have no more questions and would like to recommend the publication of this paper to Nature Communications.

Reviewer #2:

Remarks to the Author:

I would like to commend the authors on the effort they have made to improve the quality of the manuscript.

1) There is still a large number of grammatical errors that the authors need to address. I recommend thorough proofreading for their manuscript to ensure clarity as the writing tends to be confusing in a number of places.

As an example, the opening statement in the abstract should read "Complex-valued neural networks have many advantages over their real-valued counterpart. " rather than "Complex-valued neural networks have many advantages over its real-valued counterpart. ". The last line of the abstract mentions "strong learning ability", it is not clear what this means and how to quantify it. Does strong learning imply high accuracy?

In the abstract, the authors should refer to the task as "classifying nonlinear datasets" rather than "decision boundary visualization in classifying nonlinear datasets" which is not a machine learning task.

There are also numerous usages of light/lights to mean light signal/light signals, etc.

2) The discussion on the classification of nonlinear datasets in this version is helpful when comparing with the real-valued alternatives.

Otherwise, the rest of my concerns have been sufficiently addressed.

Manuscript ID:	Nature Communications manuscript NCOMMS-20-14526B
Paper title:	An Optical Neural Chip for Implementing Complex-valued Neural Network
Authors:	H. Zhang, M. Gu, X. D. Jiang, J. Thompson, H. Cai, S. Paesani, R. Santagati, A. Laing, Y. Zhang, M. H. Yung, Y. Z. Shi, F. K. Muhammad, G. Q. Lo, X. S. Luo, B. Dong, D. L. Kwong, L. C. Kwek, and A. Q. Liu

Reply to Reviewer 2

We are grateful to the Reviewer for the constructive comments and recommendation of our paper for publication. We are happy to address the final comments .

Comment 1: There is still a large number of grammatical errors that the authors need to address. I recommend thorough proofreading for their manuscript to ensure clarity as the writing tends to be confusing in a number of places. (1) As an example, the opening statement in the abstract should read "Complex-valued neural networks have many advantages over their real-valued counterpart. " rather than "Complex-valued neural networks have many advantages over its real-valued counterpart. ". (2) The last line of the abstract mentions "strong learning ability", it is not clear what this means and how to quantify it. Does strong learning imply high accuracy? (3) In the abstract, the authors should refer to the task as "classifying nonlinear datasets" rather than "decision boundary visualization in classifying nonlinear datasets" which is not a machine learning task. (4) There are also numerous usages of light/lights to mean light signal/light signals, etc.

Answer: As pointed by the reviewer, we have corrected and polished our manuscript thoroughly to make it more legible and comply with the policies and formatting requirements of Nature Communications.

- (1) The opening statement in the abstract is revised to
"Complex-valued neural networks have many advantages over their real-valued counterparts." on Page 2.
- (2) The last sentence of abstract is revised as
"Strong learning capabilities (i.e., high accuracy, fast convergence and the capability to construct nonlinear decision boundaries) are achieved by our complex-valued ONC compared to its real-valued counterpart." on Page 2.
- (3) The mentioned task by the reviewer is also rephrased to *"Classification of nonlinear datasets Circle and Spiral"* on Page 6.
- (4) The usage of light/lights are changed to light signal/light signals, on Page 6 and Page 7.

Comment 2: The discussion on the classification of nonlinear datasets in this version is helpful when comparing with the real-valued alternatives.

Answer: We are grateful to the Reviewer for their affirmation of our work.

We are grateful to the Reviewer for the constructive comments and recommendation of our paper for publication. We have corrected and polished our manuscript thoroughly to make it more legible and comply with the policies and formatting requirements of Nature Communications.